# The Role of Oxidative Stress in the Antifungal Activity of Two Mollusk Fractions on Resistant Fungal Strains

**DOI:** 10.3390/ijms26030985

**Published:** 2025-01-24

**Authors:** Lyudmila Velkova, Radoslav Abrashev, Jeny Miteva-Staleva, Vladislava Dishliyska, Aleksandar Dolashki, Boryana Spasova, Pavlina Dolashka, Maria Angelova, Ekaterina Krumova

**Affiliations:** 1Institute of Organic Chemistry with Centre of Phytochemistry, Bulgarian Academy of Sciences, Academician G. Bonchev Str., bl. 9, 1113 Sofia, Bulgaria; lyudmila.velkova@orgchm.bas.bg (L.V.); aleksandar.dolashki@orgchm.bas.bg (A.D.); or pda54@abv.bg (P.D.); 2Stephan Angeloff Institute of Microbiology, Bulgarian Academy of Sciences, Academician G. Bonchev 26, 1113 Sofia, Bulgaria; rabrashev@abv.bg (R.A.); j_m@abv.bg (J.M.-S.); vladydacheva@yahoo.com (V.D.); bkasovska@abv.bg (B.S.); mariange@microbio.bas.bg (M.A.); 3Centre of Competence “Clean Technologies for Sustainable Environment—Waters, Waste, Energy for a Circular Economy”, 1000 Sofia, Bulgaria

**Keywords:** antifungal activity, damaging action, mollusk fractions, *Penicillium*, *Aspergillus*, protein/reducing sugar leakage, oxidative stress biomarkers, antioxidant enzyme defense

## Abstract

Fungal infections are a significant global public health challenge because of their widespread occurrence, morbidity, and profound social and economic consequences. Antifungal resistance is also an increasing concern, posing a substantial risk to public health. There is a growing interest in searching for new antifungal drugs isolated from natural sources. This study aimed to evaluate the antifungal activity of novel mollusk fractions against fungal strains resistant to nystatin and amphotericin B. In addition, the role of oxidative stress in the mechanism of damage was determined. The mucus from the garden snail *Cornu aspersum* (MCa/1-20) and the hemolymph fraction from the marine snail *Rapana venosa* (HLRv/3-100) were obtained and characterized via 12% sodium dodecyl sulfate–polyacrylamide gel electrophoresis (SDS-PAGE) and mass spectrometric -analyses. The results demonstrate that the spores and biomass of both mollusk fractions have a significant fungicidal effect against *Penicillium griseofulvum*, and *Aspergillus niger*. Compared to the control group, the release of intracellular proteins and reducing sugars was significantly increased in the treated groups. The data showed increased levels of oxidative stress biomarkers (lipid peroxidation and oxidatively damaged proteins) and a downregulated antioxidant enzyme defense, corresponding to increased antifungal activity. To our knowledge, this is the first study evaluating oxidative stress as a factor in mollusk fractions’ antifungal activity.

## 1. Introduction

Drug resistance is one of the most pressing problems of the 21st century. Its extraordinary global importance is constantly increasing. The World Health Organization (WHO) considers it to be the most serious threat to public health today. The WHO’s most recent statistics indicate that 1.27 million people died due to antimicrobial resistance (AMR) in 2019, and the death of 5 million people was related to this problem. According to the predictions for 2050, about 8.7 million people will die due to AMR [1]. AMR occurs when microorganisms adapt in response to the use of drugs designed to inhibit or kill them [2]. Due to this drug resistance, antimicrobial agents lose their efficacy, rendering infections progressively challenging or unmanageable to treat. The development of simultaneous multidrug resistance (MDR) is characterized by acquired non-susceptibility to at least one agent across three or more antimicrobial groups [3,4]. Microbes exhibit several mechanisms, including inherent resistance to specific antimicrobials, genetic mutations, and acquired resistance from other species [4]. The presence of resistant microorganisms complicates treatment strategies, often requiring the use of alternative or higher doses of antimicrobials or leading to a shortage of effective treatment options.

Fungal infections are a major public health problem worldwide due to their high prevalence, morbidity, and social and economic impact. Fungi are ubiquitous and occupy all possible ecological niches. Their number is not exactly known. The most frequently reported data suggest fungal species are in the range of 1.5 to 10 million, depending on the method of calculation [5]. Based on comparative analyses with plants, the number of fungal species ranges between 2.2 and 3.8 million [6,7]. However, high-throughput DNA sequencing suggests a significantly higher number, between 11.7 and 13.2 million [8,9].

Fungi are part of the human microbiome, where they play an important role in health [10]. Three hundred species are known to cause human disease [11,12]. Various species belonging to the genera *Aspergillus*, *Mucor*, *Penicillium*, *Cladosporium*, *Fusarium*, *Candida*, *Cryptococcus*, *Histoplasma*, etc., have been reported as etiological agents of well-characterized respiratory diseases and opportunistic fungal infections [13,14]. The seriousness of fungal infections was reawakened during the COVID-19 pandemic in the form of life-threatening secondary infections requiring treatment in intensive care units [15].

The prevalence of mycoses has risen significantly over the past few decades [16]. These infections represent a growing, yet often overlooked, public health crisis, necessitating immediate intervention to avert further deterioration [17]. Concurrently, there is a notable increase in antifungal resistance. This resistance can develop either through the selective pressure exerted by antifungal treatments in individual patients or via the horizontal transfer of resistant strains among patients [18,19]. As a result, traditional antifungal medications are becoming less effective, which contributes to higher mortality rates. This situation is exacerbated by the limited availability of new antifungal agents. A primary obstacle arises from the shared eukaryotic nature of fungi and their human hosts, which complicates the identification of distinct therapeutic targets.

The development of antifungal medications is increasingly centered on identifying alternative agents that demonstrate high efficacy, minimal resistance, limited side effects, and synergistic antifungal properties, as well as those derived from natural sources [20,21]. Recent studies have underscored the potential of proteins and peptides extracted from various mollusks to inhibit microbial proliferation. The majority of investigations have focused on their antibacterial properties [22,23,24,25]. For instance, Brakemi et al. [26] conducted a comprehensive review of the antimicrobial effects of extracts obtained from natural mollusk shells. Notably, alpha-helical peptides derived from the mollusk *Pomacea poeyana* exhibited significant antibacterial activity against *Pseudomonas aeruginosa* [27].

The inhibition of fungal growth is relatively less studied. The soft tissue extracts from *Helix aspersa* exerted antifungal activity against *Candida albicans*, *Aspergillus flavus*, and *Aspergillus brasiliensis* in a concentration-dependent manner [28]. Similarly, peptides isolated from the marine mollusk *Nerita versicolor* demonstrated antibiofilm action against several *Candida* species [29]. Crude extracts of 18 molluscan species were reported to be effective antifungals against strains belonging to the genera *Penicillium*, *Trichoderma*, *Trichothecium*, *Hormodendrum*, and *Rhizophus* [30].

Research indicates that extracts derived from mollusks exhibit significant activity against both bacterial and fungal pathogens. Specifically, the mucus produced by the marine mollusk *Melo melo* contains elements that demonstrate strong antibacterial effects against *Klebsiella pneumoniae* and *Salmonella typhi*, as well as antifungal properties against *Trichophyton mentagrophytes* and *A. flavus* [31]. In a study of seven different snail species, the protein extracted from *Cryptozona bistrialis* exhibited the highest level of antimicrobial activity against various pathogenic bacteria and fungi.

In recent years, there has been a growing interest in natural antifungal compounds, especially concerning their mode of action [32,33,34,35]. Li et al. [36] have compiled a detailed summary of the mechanisms employed by antifungal peptides, which vary according to their pathways of action. Their antifungal activity could involve several mechanisms, such as oxidative damage, osmotic stress, apoptosis, the disruption of cytoskeletal integrity, and metabolic dysfunction within cells. These compounds can affect cell walls, membranes, nucleic acids, organelles, and other intracellular macromolecules. Typically, antifungal agents operate by either directly killing the pathogens or by inhibiting their growth and reproduction.

Recent years have witnessed an increase in research interest in the impact of oxidative stress (OS) on the efficacy of antifungals [37,38,39,40,41]. This stress is a consequence of the production of reactive oxygen species (ROS) during aerobic cellular metabolism. ROS include a variety of radical and non-radical molecular species, such as singlet oxygen (^1^O_2_), the superoxide radical anion (^•^O_2_^−^), hydrogen peroxide (H_2_O_2_), and the hydroxyl radical (HO^•^), along with other harmful chemical agents. These reactive species can lead to significant chemical modifications in proteins, lipids, polysaccharides, DNA, RNA, and small metabolites, resulting in cellular damage and potential cell death [42]. The antifungal activities observed in plant extracts [43,44], essential oils [41], and peptides from diverse natural sources, including microorganisms and invertebrates [36], are largely attributed to OS. However, research on the antifungal activity of peptides and fractions from mollusks remains limited. Furthermore, there is a lack of comprehensive information regarding the role of oxidative stress in their fungicidal effects. Despite the scarcity of studies on the damage mechanisms of mollusk fractions, it is important to note that considerable diversity exists and the mechanisms depend on the specific fungicide and the taxonomic classification of the strains being treated [29,32,45].

Our earlier investigations have shown that a diverse array of new protein and peptide fractions extracted from marine and terrestrial mollusks possess significant antifungal properties [46]. The fractions that exhibited the highest efficacy were identified as the mucus fraction, with a molecular weight ranging from 1 to 20 kDa, from the garden snail *Cornu aspersum* (MCa/1-20), and the hemolymph fraction with a molecular weight of 3–100 kDa, from the marine snail *Rapana venosa* (HLRv/3-100). Both fractions effectively suppressed the growth of fungal strains belonging to the *genera Aspergillus*, *Mucor*, *Penicillium*, *Cladosporium*, *Fusarium*, *Alternaria*, *Botrytis*, and *Candida*. It is essential to explore the impact of these fractions on fungal strains that are resistant to existing commercial agents.

The present study aimed to determine the ability of two novel mollusk fractions (MCa/1-20 and HLRv/3-100) to inhibit the growth of conidiospores and mycelia of three fungal strains (*Aspergillus niger*, *Penicillium griseofulvum*, and *Mucor michei*) resistant to nystatin (Nys) and amphotericin B (AmB). Furthermore, studying the oxidative stress response at sublethal concentrations provided new information about the role of OS in the damage mechanism of both fractions.

## 2. Results

### 2.1. Characterization of the Mucus Fraction with an MW of 1–20 kDa via Matrix-Assisted Laser Desorption/Ionization Time-of-Flight Mass Spectrometry (MALDI-Tof-MS) Analyses

The fraction with an MW of 1–20 kDa from the *C. aspersum* mucus was obtained by concentrating the crude extract fraction with an MW of less than 20 kDa, resulting in a concentration of 0.95 mg/mL. The peptides in this fraction were characterized via mass spectrometric analyses using AutoflexTM III, High-Performance MALDI-TOF, and TOF/TOF systems (Bruker Daltonics, Bremen, Germany). Recently, a selection of peptides with MW < 3 kDa was characterized, focusing on their molecular masses and primary structures (determined via MS-and MS/MS analyses). Furthermore, several physicochemical parameters, such as isoelectric points (pI), the grand average of hydropathicity (GRAVY), net charge, and potential antimicrobial activity (predicted by iAMPpred software) were evaluated [47]. The results reveal a variety of cations, anions, and neutral peptides, primarily characterized by a hydrophobic surface, with potential antibacterial, antifungal, and antiviral activities, as presented in Table 1.

The MALDI-MS spectrum, recorded in the range from 3 to 20 kDa (Figure 1), shows that the peptides defined as protonated molecule ions [M + H]^+^ at *m*/*z* 4101.24 Da, 4498.48 Da, 6371.31 Da, 7829.75 Da, and 10,442.46 Da are dominant (Figure 1). The ions [M + H]^+^ at *m*/*z* 13,218.49 Da, 15,058.77 Da, and 18,001.26, although of lower intensity, are also clearly presented (Figure 1). These results are in agreement with recently published research on mucus fractions with an MW < 20 kDa from *C. aspersum* [47].

### 2.2. Characterization of the Hemolymph Fraction with an MW 3–100 kDa via Electrophoretic Analyses

The fraction derived from the hemolymph of *R. venosa*, characterized using molecular weights ranging from 3 to 100 kDa, was analyzed using 12% SDS-PAGE and evaluated with ImageQuant^TM^ TL v8.2.0 software. The electrophoretic profile presented in Figure 2a,c clearly indicates the presence of various protein bands, primarily located within the 10 to 100 kDa molecular weight range. The highest expression was observed for proteins with MWs of 49.293 kDa and 37.851 kDa, followed by proteins with an MW of 93.765 kDa, 62.601 kDa, and 26.191 kDa, while proteins with an MW of 21.247 kDa, 17.655 kDa, and 12.942 kDa had significantly lower expression. The results obtained are consistent with the previously identified proteins in the 50–100 kDa range of the *R. venosa* hemolymph fractions, which were recognized for their antibacterial and antitumor activities [52,53].

The protein band with the highest expression at 49.293 kDa (Figure 2) most probably included functional units (FUs) of *R. venosa* henocyanin derived from endogenous proteolytic processes. This hypothesis was recently confirmed through the identification of proteins in a fraction with an MW of 50–100 kDa, obtained from the *R. venosa* hemolymph [52,53].

Based on the search conducted in the database UniProt (https://www.uniprot.org) for proteins localized extracellularly in Gastropoda, we propose that the protein band with a high expression level at 37.851 kDa may present the extracellular protein cathepsin-L endopeptidase. This protein is present in the hemolymph of abalones, including *Haliotis discus discus*, which has a theoretical MW of 38.963 kDa (UniProt ID A0A1P8DD91), and *Haliotis discus hannai*, with an MW of 36.305 kDa (UniProt ID K7QTY9). We also considered the Cathepsin L-like cysteine proteinase identified in the hemolymph of *Haliotis diversicolor* supertexta, which has a molecular weight of 39.220 kDa (UniProt ID C5IIM1). These proteins belong to the Cathepsin family and play an important role in the innate immunity of the invertebrate organisms [54].

The proteins detected at the electrophoretic bands of 93.765 kDa and 62.601 kDa were recently characterized as peroxidase-like proteins and a protein with L-amino-acid oxidase (LAAO) activity, respectively [52,53]. The peroxidase-like protein was found in the hemolymph *Lottia gigantea* (UniProt ID B3A0P3, with a theoretical MW of 92.943 kDa). The proteins with L-amino-acid oxidase activity were identified in the hemolymph of *Aplysia californica* (Uniprot ID: Q6IWZ0, MW 60.300 kDa) and Aplysianin A in *Aplysia kurodai* (Uniprot ID: Q17043, MW 62.376 kDa).

Other extracellular proteins, such as galectins and lectins, are also contained in the hemolymph of various gastropods, mollusks, and marine arthropods. In this context, the protein expressed at 17.655 kDa could correspond to galectins with theoretical MWs of 16.480 kDa and 17.452 kDa, identified in the hemolymph of sea slug *Elysia chlorotica* (with UniProt IDs of A0A433TBR5 and A0AAE0XNA, respectively). Furthermore, galectins with theoretical molecular weights of 16.636 kDa and 17.166 kDa have been reported in the freshwater snail *Biomphalaria pfeifferi* (UniProt IDs of A0AAD8C3C5 and A0A9W3ABA3), along with a lectin exhibiting a theoretical MW of 16.558 kDa from *Haliotis rufescens* (California red abalone, UniProt ID A0A0A7HIM4).

The protein band detected at 26.191 kDa is likely to contain a variety of lectins, particularly C-type and H-type lectins, along with galectins. An examination of the UniProt database indicates several C-type lectins with theoretical molecular weights between 25 and 27.5 kDa. Examples include C-type lectin domain-containing proteins from *E. crispata* (MW 25.857 kDa, UniProt ID A0AAE0ZY08), *L. gigantea* (MW 27.170 kDa, UniProt ID V4BQI3), and a Perlucin-like protein in the hemolymph of *B. glabrata* (MW 25.616 kDa, UniProt ID A0A9W3BPT7). Furthermore, proteins resembling galectins with theoretical molecular weights of 26.006 kDa and 24.392 kDa were identified in *E. chlorotica* (UniProt ID A0A3S0ZDA4 and UniProt ID A0A433SIR7), as well as a 26.371 kDa protein in the hemolymph of *B. pfeifferi* (UniProt ID A0A2C9LC52).

### 2.3. Antifungal Activity of Mollusk Fractions Compared to Nys and AmB

A preliminary screening of mollusk extracts’ antifungal activity at different concentrations revealed that the MCa/1-20 fraction and the hemolymph fraction of HLRv/3-100 exhibited a significant inhibitory effect against three fungal strains—*A. niger*, *P. griseofulvum*, and *M. michei*—all of which were found to be resistant to Nys and AmB.

The data presented in Table 2 show the MICs of all antifungal agents tested against the development of spores of the resistant strains mentioned above. The mollusk extracts exhibited remarkably similar MIC values. They possess a high antifungal potential that is comparable to that of the commercial drugs Nys and AmB. It should be noted that the inhibitory effect is strain-dependent. The MIC values for MCa/1-20 against *A. niger*, *P. griseofulvum*, and *M. michei* were 1.75, 1.75, and 3.50 g/mL, respectively. The results for HLRv/3-100 were similar, with respective measurements of 3.50, 1.75, and 3.50 µg/mL.

In comparison, the MICs of AmB, ranging from 4 to 8 µg/mL, were found to be greater than those for MCa/1-20 and HLRv/3-100, indicating that AmB exhibits lower in vitro activity compared to these fractions. Additionally, the MIC measurement revealed that the antifungal efficacy of Nsy was higher than that of AmB. However, MCa/1-20 and HLRv/3-100 proved to be more effective inhibitors of *A. niger* and *P. griseofulvum*. Conversely, for *M. michei*, the mollusk fractions displayed reduced in vitro activity compared to Nys.

According to the findings from the MIC evaluations, MCa/1-20 and HLRv/3-100 had the strongest antifungal effect against *A. niger* and *P. griseofulvum*, followed by Nys and AmB. Concerning the test strain *M. michei*, Nys showed greater in vitro efficacy than MCa/1-20, HLRv/3-100, and AmB.

### 2.4. Inhibitory Effect on Fungal Growth Under Submerged Cultivation

In the preliminary studies, concentrations of 0.5, 1.0, and 3% HLRv/3-100 or MCa/1-20 were used. The data showed the complete inhibition of spore development at 1 and 3% concentrations and a strong retardation of development at a 0.5% concentration. Following this, subsequent experiments were conducted using lower concentrations of 0.01%, 0.05%, 0.1%, 0.3%, and 0.5%. The application of both mollusk fractions at the above-mentioned concentrations resulted in a clear decrease in biomass content (Figure 3). In the case of *A. niger*, spore development and mycelial growth were evident at concentrations of 0.01, 0.05, and 0.1% for MCa/1-20. However, the higher concentrations of 0.3 and 0.5% completely inhibited this process, indicating a pronounced fungicidal effect. This strain demonstrated a slightly higher resistance to HLRv/3-100, showing limited growth at 0.3% and fungicidal properties at 0.5%. Similarly, *P. griseofulvum* showed a parallel trend when subjected to MCa/1-20 or HLRv/3-100, achieving complete inhibition at 0.5%.

### 2.5. Protein and Reducing Sugar Leakage of A. niger and P. griseofulvum

The antifungal effect of the mollusk fractions against *A. niger* and *P. griseofulvum* was assessed by measuring the extracellular content of soluble protein and soluble sugar (Figure 4).

The results revealed a dose-dependent enhancement in the leakage of both intracellular substances after 6 h exposure to the tested fractions. The cell responses of *A. niger* to MCa/1-20 and HLRv/3-100 were very similar, with a gradual increase in protein and sugar leakage corresponding to increasing concentrations (Figure 4A,C). Extreme escape was evaluated at concentrations of 0.1 and 0.3%, which correspond to the MIC values. Following treatment with MCa/1-20, the protein and sugar levels were found to be four and six times greater than those in the control group, respectively. Comparable results were found regarding the impact of HLRv/3-100 on *A. niger*, with peak protein and sugar leakage occurring at the same concentrations of 0.1% and 0.3%.

The results for *P. griseofulvum* were similar to those for *A. niger*, except that significant leakage occurred for P. griseofulvum at a concentration of 0.05% (Figure 4B,D). At 0.05%, 0.1%, and 0.3%, respectively, the extracellular component content increased 3.5-, 5-, and 6.4-fold in comparison to the control.

### 2.6. Oxidative Stress Induction by Mollusk Fractions MCa/1-20 and HLRv/3-100

The present study showed a complete inhibition of fungal spore development at 0.3 and 0.5%. To explore the correlation between antifungal efficacy and oxidative stress, it is essential to utilize enough mycelial. Therefore, a condition of partial growth inhibition is required. In the subsequent experiments, concentrations of 0.05% and 0.1% were applied. The evaluation of potential oxidative stress in cultures of *A. niger* and *P. griseofulvum* subjected to the mollusk fraction included the measurement of biomass content, along with the assessment of oxidatively damaged proteins and malondialdehyde (MDA) levels (Figure 5).

According to the results of the oxidative stress experiments, the mollusk fractions inhibited the mycelial growth of both strains compared to the control. However, the concentrations that were applied allowed for the formation of the desired biomass content (Figure 5A,B). The degree of inhibition is more closely related to the strain than to the type of fraction used. The application of MCa/1-20 and HLRv/3-100 resulted in a 2- and 3-fold reduction in A. *niger* growth, respectively. However, their efficacy against *P. griseofulvum* was somewhat lower, with 1.5 and 2-fold reductions in growth compared to untreated cultures.

The formation of a carbonyl group is a marker of protein oxidation. These groups arise in polypeptide chains during the reaction of proteins with ROS [55]. Notably, there was a significant increase in carbonyl group levels in the treated cultures of *A. niger* and *P. griseofulvum* compared to the untreated cultures. The results showed changes in the level of oxidatively damaged proteins induced by the MCa/1-20 fraction that were comparable to those caused by the HLRv/3-100 fraction (Figure 5C,D). These data suggest that the mollusk fractions operate via a comparable damaging mechanism. Furthermore, the proteins from both strains exhibited comparable susceptibility to the antifungal agents used.

To evaluate the oxidative status of cell membranes, lipid peroxidation was measured using MDA content as a biomarker. Figure 5E,F illustrate the significant increases in the MDA levels of all treated groups compared to the control group. Specifically, the treatment of *A. niger* with MCa/1-20 and HLRv/3-100 at concentrations of 0.05 and 0.1% resulted in MDA levels that were approximately 2- and 3-fold higher than those of the control. The strain *P. griseofulvum* displayed a comparable response.

### 2.7. Antioxidant Enzyme Response to the Mollusk Fractions

The highly increased levels of oxidative stress biomarkers inside the treated fungal cells suggest that oxidative stress was induced. To confirm this suggestion, the activities of two antioxidant enzymes (SOD and CAT) involved in the intracellular defense were determined.

SOD activity in both tested strains showed a significant increase following treatment at a concentration of 0.05%, with comparable effects observed for MCa/1-20 and HLRv/3-100 (Figure 6A,B). In the case of *A. niger*, treatment with MCa/1-20 resulted in a 2.7-fold increase in SOD activity, while HLRv/3-100 led to a 3.4-fold increase (Figure 6A). At the same time, cultures of *P. griseofulvum* subjected to 0.05% MCa/1-20 showed a 2.1-fold enhancement in SOD activity, whereas treatment with HLRv/3-100 resulted in a 1.7-fold increase (Figure 6B). In contrast, the SOD levels induced by a 0.1% concentration were lower than those observed with the 0.05% treatment but still exceeded the control levels.

Cultures of *A. niger* and *P. griseofulvum* exposed to MCa/1-20 showed a dose-dependent reduction in CAT activity compared to the control group (Figure 6C,D). Correspondingly, a slight increase in CAT activity was observed in cells treated with 0.05% HLRv/3-100, which was followed by a decline in activity at a concentration of 0.1% relative to the control samples.

## 3. Discussion

The increasing resistance of fungal strains to existing antimycotics is a major global problem. An effective solution is the development of new molecules with antifungal activity. Mollusk fractions are promising options because of their broad antimicrobial activity and their ability to impede the development of resistance [29]. In the present study, we demonstrated that the MCa/1-20 fraction from garden snail *C. aspersum* and HLRv/3-100 fraction from marine snail *R. venosa* possess effective antifungal properties against Nys- and AmB-resistant strains of *A. niger*, *P. griseofulvum*, and *M. michei*.

### 3.1. Antifungal Effect of the Mollusk Fractions MCa/1-20 and HLRv/3-100

The first finding is the significant antifungal activity of both fractions. According to the MIC values, MCa/1-20 and HLRv/3-100 inhibited spore development more potently than AmB and Nys. (Table 2). Furthermore, the antifungal activity of both fractions was also confirmed in submerged cultures (Figure 3). The effect was dose-dependent. The antifungal potential of mollusk fractions has not been extensively studied in comparison to commercially available agents. In our previous study [46], the *R. venosa* fraction was found to prevent the growth of *A. solani*, *M. hiemalis*, and *B. cinerea* more effectively than the antifungal drug nystatin, confirming the current results. In terms of fungicidal activity, extracts from mollusk shells and the sea sponge *Euryspongia* sp. were found to be similar to those of AmB and Nys [56,57]. According to Gutierrez-Gongora et al. [45], two freshwater mussel species showed an inhibitory effect against a fluconazole-resistant strain of *C. neoformans*. Compared to fluconazole, soft tissue extracts from *H. aspersa* also showed stronger antifungal activity against *C. albicans*, *A. flavus*, and *A. brasiliensis* [28].

Although both fractions show similar antifungal activity, they are completely different in their composition. The established antifungal activity of MCa/1-20 is related to the presence of antimicrobial peptides with MW 1–3 kDa, as well as peptides with higher molecular masses, polypeptides, and proteins with low molecular weight. The peptides shown in Table 1 contain high levels of glycine, leucine, valine, and proline amino acid residues, as well as the presence of tryptophan, aspartic acid, phenylalanine, and arginine, which are associated with their antimicrobial activity. Furthermore, they show high homology with known antimicrobial peptides [47]. The peptides with the highest prognostic antifungal activity are Nos. 3, 7, 8, 13, 17, 18, and 21–23 (Table 1). The determined ions [M + H]^+^ at *m*/*z* 9269.03 Da, 10,442.46 Da, 13,218.49 Da, 15,058.77 Da, and 18,001.26 Da are in good agreement with similar proteins in the snail mucus of *C. aspersum*, as reported previously [47,50]. For example, the peptide detected as [M + H]^+^ at *m*/*z* 9269.03 Da is in good agreement with the AMP named mytimacin-AF (9700 Da) in the mucus of *Achatina fulica*, which exhibits antimicrobial activity against *S. aureus*. *Bacillis* spp., *K. pneumoniae*, and *C. albicans* [58]. The ion determined at 13,218.49 Da agrees well with the glycosylated monomers of *H. aspersa* agglutinin (HAA), which are H-lectins, determined at *m*/*z* 13,046 Da [59]. The protein detected at *m*/*z* 18,001.26 Da (Figure 1) probably corresponds to the anti-*P. aeruginosa* protein (MW ~17.5 kDa) described in a study by Pitt et al. [60].

Several studies have shown that proteins in the hemolymph of Mollusca are mainly different forms of hemocyanin, protease inhibitors, alpha-2-macroglobulin, a putative clotting protein, actin, and many others [61]. Data on proteins other than the hemocyanin in the hemolymph of Gastropoda are still limited. The extracellular protein composition of *R. venosa* hemolymph, until recently, mainly included some functional units of *R. venosa* hemocyanin, identified via their amino acid sequences and carbohydrate structures, antimicrobial proline-rich peptides, and cytoplasmic actin, determined using its gene sequence [62,63]. These proteins show high homology with peroxidase-like protein (*L. gigantea*), aplicyanin A (*A. kurodai*), and L-amino-acid oxidase LAAO (*A. californica*), as well as FUs, with MWs of about 50 kDa in *R. venous* hemocyanin, as determined via proteomic analysis. These proteins were also detected in the HLRv/3-100 fraction (Figure 2), which suggests significant antifungal activity.

We assumed that the synergy between bioactive compounds in the hemolymph fraction is responsible for the observed antifungal activity. The protein band with the highest expression included hemocyanin functional units with an MW of around 50 kDa, identified via proteomic analysis by Kirilova et al. [52] and Petrova et al. [53]. A previous study reported the antibacterial activity of hemocyanins in *H. aspersa* and *R. venosa* against *Staphylococcus aureus*, *Streptococcus epidermidis*, and *E. coli* [64]. The protein cathepsin-L, as well as the cathepsin-L-like cysteine proteinase (putative proteins on a band with an MW of 37.851 kDa, as shown in Figure 2a,c), are present in the hemolymph of arthropods and gastropods. It is known that cathepsins can directly interact and participate in the destruction of invading bacteria, as well as promoting phagocytosis and contributing to the stimulation of the protective microbial-specific immune response by regulating the processing and presentation of bacterial antigen [65,66]. Cysteine proteases are found in all living organisms and are involved in various biological processes, including immune functions. To date, few cysteine proteases have been characterized in mollusks; these are thought to be involved in innate immune responses in these species [54]. We assume that extracellular endopeptidases such as cathepsin L and cathepsin L-like cysteine proteinase, detected in the hemolymph of the same gastropods with an MW of 36–39 kDa, may also be associated with antifungal activity, as they play an important role in the innate immunity of invertebrate organisms.

The protein detected at 93.765 kDa (Figure 2) corresponds to a peroxidase-like protein that is involved in the oxidative stress response, catalyzing the oxidation of many aromatic amines and phenols via hydrogen peroxide [67]. Some peroxidases exhibit antifungal activity [68,69]. The observed protein expression with an MW of 62.601 kDa corresponds to that of proteins with L-amino acid oxidase (L-AAO) activity, which are also widespread in 16 species of gastropods [52,53]. The presence of a protein with L-AAO activity in the protein band at 62.601 kDa in the HRv 3/100 kDa may explain not only the observed antibacterial activity against *E. coli* [52] but also the high antifungal activity of the fraction with an MW of 50–100 kDa against six pathogenic fungal strains [46].

Galectins and lectins, found in the hemolymph of many marine and freshwater organisms (belonging to Mollusca and Arthropoda), are multifunctional proteins with agglutination activity. Galectins are a family of lectins characterized by their binding affinity for beta-galactosides. Their carbohydrate-recognition domain can interact with both pathogens and parasites, promoting their phagocytosis [61]. Moreover, for invertebrates, C-type lectins play an extremely important role in the innate immune system, protecting against invading microorganisms via phagocytosis, cellular encapsulation, and agglutination [70,71]. We assume that the proteins corresponding to the electrophoretic bands at 17.655 kDa and 26.191 kDa belong to these proteins and therefore may also be related to the acquired antifungal activity of the HLRv/3-100 fraction.

Antimicrobial proline-rich peptides with an MW between 3.0 and 9.5 kDa, identified in the *R. venosa* hemolymph by Dolashka et al. [62], probably also contributed to the observed antifungal activity. Four of the Pro-rich peptides showed strong antimicrobial activity against tested Gram-positive and Gram-negative bacteria [63].

### 3.2. Effect of the Mollusk Fractions MCa/1-20 and HLRv/3-100 on the Fungal Cell Membranes

The treatment of *A. niger* and *P. griseofulvum* with the mollusk fractions induced changes in the extracellular content of both proteins and reducing sugars (Figure 4). Compared to the control group, the release of the intracellular components through the cell membrane was significantly increased in the treated groups. This effect was positively correlated with the concentration of used fractions. The most leakage was observed at concentrations of 0.1 and 0.3%. Protein and sugar leakages are undoubtedly indicative of membrane damage. Therefore, the MCa/1-20 and HLRv/3-100 fractions can compromise the integrity of the cell membrane in both fungal cultures and enter the cells. This may lead to the disturbance of multiple cellular functions.

Our results are comparable to those obtained for cell-free supernatants from *Lactobacillus pentosus* [72], volatile organic compounds synthesized by endophytic fungal isolates of garden nasturtium [73], methanolic extracts derived from *Myrtus communis* [34], the inflorescence extract of *Euphorbia hirta* L. [74], and many more. All of these investigations demonstrated that antifungal agents can damage microbial cell membranes. The lethal leakage of biologically valuable macromolecules, i.e., proteins, sugars, enzymes, K^+^ ions, DNA, and so forth, causes a change in the permeability of cell membranes, which in turn inhibits fungal growth.

However, studies using mollusk extracts are extremely rare. For example, silver nanoparticles in the mucus of the land snail *A. fulica* showed antimicrobial activity, with the simultaneous release of protein and reducing sugars [75]. As far as we know, the present study is the first to investigate the leakage of intracellular components through the fungal cell membrane after treatment with mollusk fractions.

### 3.3. Induction of OS in the Fungal Cells Following Treatment with Mollusk Fractions

#### 3.3.1. Changes in Biomarkers of OS

The main finding of the present results is an increased level of oxidative stress in the presence of mollusk fractions. As is known, the overproduction of ROS damages DNA, proteins, and lipids in cells, causing irreversible oxidative damage [40,42]. The sublethal doses of the tested fractions MCa/1-20 and HLRv/3-100 allowed for the *A. niger* and *P. griseofulvum* stains to grow under submerged conditions (Figure 5A,B). Despite the decrease in biomass content, the treated variants accumulated enough mycelium to evaluate the oxidatively damaged protein and lipid peroxidation.

As expected, the treatment of *A. niger* and *P. griseofulvum* with the tested mollusk fractions led to a substantial increase in the MDA levels in 36 h mycelium in comparison to the control (Figure 5). MDA has been identified as a reliable marker of oxidative stress-mediated lipid peroxidation. Furthermore, lipid peroxidation is recognized as an important indicator of ROS-induced oxidative stress, which rapidly generates polar lipid hydroperoxides through reactions with unsaturated lipids [32,40]. Thus, it can be assumed that the MCa/1-20 and HLRv/3-100 fractions induced an increase in the level of oxidative stress in the cells of the tested strains. The present results provide evidence that the investigated extracts changed the lipid bilayer of the fungal membranes, thereby establishing a potential mechanism of damage. They point to a process involving cellular fatty acid double bonds and lipid peroxidation. The positive correlation between the increased MDA content and the antifungal effect of the used fractions was also observed after treatment with other natural compounds, such as geraniol and o-vanillin [39,76], extracts from endophytic *A. terreus* [77], essential oils [32,78], etc. In contrast, cinnamaldehyde significantly reduced the level of lipide peroxidation in *A. flavus*, suggesting that its inhibitory effect may be related to the reduction in oxidative stress caused by changes in cellular structure [79].

The results of the present study also showed a remarkable concentration-dependent increase in protein carbonyls as a result of treatment with the mollusk fractions MCa/1-20 and HLRv/3-100 compared to the control (Figure 6). This upward trend was observed in both *A. niger* and *P. griseofulvum* strains. Protein carbonylation is one of the most common oxidative modifications [80]. While many different types of oxidative protein modifications are possible, the assay of protein carbonyls (aldehydes and ketones) is the most frequently employed. Carbonylated proteins occur relatively early and are quite stable; therefore, employing protein CO groups as oxidative stress indicators has various advantages over assessing other oxidized products. Protein oxidation is a particular concern since it can result in aggregation, polymerization, unfolding, or conformational changes that may impair structural or functional activity. Oxidized protein aggregates are not readily degraded in the cell, and their accumulation causes cell dysfunction [40].

Based on the results of significantly increased levels of lipid peroxidation and oxidatively damaged proteins, it could be suggested that the mollusk fraction increases ROS levels in the treated cells. The same correlation between oxidatively damaged intracellular molecules (lipids and proteins) and the induction of ROS generation was reported regarding the fungicidal activity of several natural extracts and commercial drugs. [8,39,40,81]. Khan et al. [32] further proposed that the antifungal activity of methyl eugenol, eugenol, and estragole against *C. albicans* is due to the enhancement of oxidative stress. According to reports, the antifungal activity of various agents (including thymol, farnesol, citral, nerol, and pyocyanin) against *A. fumigatus* is mediated through ROS generation [82]. Shen et al. [83] reported that thymol facilitates ROS generation in the spores of *A. flavus*, which contributes to its fungicidal activity. Several papers have suggested that the ROS generated after fungicide treatment are of mitochondrial origin [84,85,86]. Zhu et al. [87] demonstrated that the inhibition of mitochondrial function using specific inhibitors or genetic manipulation led to a decrease in ROS levels following treatment with amphotericin B and itraconazole.

#### 3.3.2. Antioxidant Response Against Both Mollusk Fractions

Interestingly, the results of the present study demonstrated both an increase in the degree of oxidative stress and a decrease in antioxidant defense when higher concentrations of the fractions were used. While higher SOD activity was linked to increased antifungal activity at a fraction concentration of 0.05% compared to the control, treatment with a higher concentration (0.1%) was linked to a decrease in enzyme activity. In addition, CAT activity was downregulated despite the higher concentration and increased antifungal activity. Comparable results were reported for other antifungal agents. For example, the activities of important enzymes responsible for detoxifying cellular ROS, CAT, and SOD, exhibited a decrease following an increase in ROS production induced by itraconazole in *C. parapsilosis* [86]. Research conducted by de Nollin et al. [88] demonstrated that miconazole enhances catalase activity in *S. cerevisiae* and *C. albicans* at fungistatic concentrations, but this activity is inhibited at fungicidal doses. Similar findings were reported for amphotericin B, fluconazole, and ketoconazole in their use against *S. cerevisiae* and *C. albicans* [89]. The authors postulated that yeast cells’ stress response and increased mitochondrial activity cause an excess of harmful ROS, which in turn causes cell death. Martins et al. [90] support these findings for the use of sub-MIC miconazole concentrations in the treatment of *S. cerevisiae*. A decrease in antioxidant enzyme activity was reported as an antifungal effect of naturally occurring plant compounds, e.g., fractions from *Lawsonia inermis* against *A. niger* and *Fusarium oxysporum* [91], *Satureja khuzistanica* Jamzad against *F. solani* K [92], and *Jacquinia macrocarpa* against *F. verticillioides* [93]. Under low-level ROS production, the cells can upregulate the antioxidant enzyme defense. However, if ROS production goes above a certain threshold, the excessive activation of stress responses is unable to protect the cells but can instead cause cellular death by depleting ATP and generating ROS [89,94]. Moreover, the decrease in CAT activity can be ascribed to the cytotoxic impacts of accumulating ROS, which substantially contribute to this phenomenon [95]. As is known, the accelerated generation of superoxide radicals leads to the inhibition of catalase activity [96]. Antifungal fractions may also suppress transcriptional regulators or act directly on genes to prevent biosynthesis and/or metabolism [95]. According to Gonzalez-Jimenez et al. [40], the ROS produced by antifungal agents can have a wide range of effects on cells, which require further investigation.

It is important to note that some antimicrobials, including antifungal compounds, can display both antioxidant and prooxidant properties depending on the concentration used, the cell type, the exposure time, and the environmental conditions [93]. Prooxidant activity is demonstrated when the treatment uses higher concentrations [97]. Moreover, prooxidant activity is more commonly displayed by compounds with smaller molecules than by those with larger molecules [98,99,100].

#### 3.3.3. Role of the OS in the Mode of Action of Mollusk Fractions

Taken together, the present data suggest the involvement of OS in the antifungal activity of a low molecular fraction derived from garden snail *C. aspersum* (MCa/1-20) and marine snail *R. venosa* (HLRv/3-100) against *A. niger* and *P. griseofulvum*. Our results clearly showed that the enhanced level of oxidative stress and the decrease in antioxidant enzyme levels in the fungal mycelia of *A. niger* and *P. griseofulvum* corresponded to increased antifungal activity.

It should be noted that, in addition to the OS, numerous important effects were reported as detrimental mechanisms of natural products or pharmaceuticals. Antifungal agents can damage mitochondria, efflux pumps, and membranes, as well as inhibiting DNA and RNA, protein synthesis, and ergosterol synthesis [35,101]. However, the major role of OS in the mechanism of toxicity of various fungicides has been confirmed by a range of studies [8,32,36,39,102]. ROS generation is regarded as one of the main biochemical reasons for apoptosis [40,81]. The OS generated by the fungicidal drugs AMB, miconazole, and ciclopirox olamine causes oxidative injuries in the cells of *C. albicans* and *S. cerevisiae* [89]. The study demonstrated that DNA damage plays a pivotal role in antifungal-induced cellular death. Moreover, the observed increase in fungicidal efficacy was linked to the suppression of DNA repair mechanisms. Khan et al. [38] suggested that the inhibition of growth and cell death that occurred in *A. nidulans* were due to the increased levels of oxidative stress and enhanced damage to the cellular membranes.

According to published reports, OS has been linked to inhibitions in the growth of treated fungal cultures. This suggests that ROS-induced damage and the antioxidant defense response could be an important target for antifungal drugs [103] and form the basis of an effective strategy for the development of new drugs [104]. In this context, the results obtained for the MCa/1-20 and HLRv/3-100 fractions may be useful in the development of antifungal drugs. In addition, our unpublished data demonstrate their safety for humans [105]. An evaluation of the cytotoxicity of the mucus fraction on the viability of non-tumorigenic dermal fibroblasts (BJ) and human keratinocytes (HaCaT) over a wide concentration range, from 1.5 to 480.0 µg/mL, for 24 and 48 h clearly showed a lack of cytopathic effect (preliminary data, [105]).

## 4. Materials and Methods

### 4.1. Materials

All chemicals were purchased from commercial suppliers and used as received. Amphotericin (AmB) and Nystatin (Nys) were purchased from Sigma-Aldrich (Merck KGaA, Darmstadt, Germany).

### 4.2. Methods

#### 4.2.1. Preparation of the Mollusk Fractions

The fractions used in the present study were obtained via ultrafiltration of the hemolymph of the Black Sea snail *R. venosa* and the mucus of the garden snail *C. aspersum*. *R. venosa* hemolymph was collected from a small incision in the soleus muscles. After the removal of gross impurities, hemocytes, and other cells via centrifugation at 10,000× *g* rpm for 20 min, the resulting supernatant was filtered and ultra-filtrated on a 100 kDa membrane (Millipore™ Ultrafiltration Membrane Filters, Regenerated cellulose, Burlington, MA, USA). The two main fractions that were obtained contained bioactive components with MWs above and below 100 kDa. The fraction including compounds with an MW below 100 kDa was concentrated via ultrafiltration on a 3 kDa membrane. In this way, the hemolymph fraction with an MW of 3–10 kDa (HLRv/3-100) was obtained, with a protein concentration of 1.42 mg/mL (according to Bradford assay) [106].

The native mucus was collected from snails of the genera *C. aspersum*, grown in Bulgarian eco-farms according to a patented technology, as described previously [107,108]. The resulting crude mucus extract was purified through several centrifugation and filtration steps [49,108]. The purified native mucus extract was separated into two major fractions via ultrafiltration under pressure with polyethersulfone membrane filters with pore sizes of 20 kDa (Microdyn Nadir™ from the STERLITECH Corporation, Goleta, CA, USA). The investigated mucus fraction MCa/1-20 was obtained via additional ultrafiltration of the fraction with an MW < 20 on Amicon Stirred Cell with a 1 kDa membrane (1 kDa NMWL, Ultracel^®^ regenerated cellulose, Millipore, Burlington, MA, USA) with a concentration of 0.95 mg/mL (according to the Bradford method [106]).

#### 4.2.2. SDS-PAGE Analysis of the Fraction from *R. venosa* Hemolymph (HLRv/3-100)

The fraction from *R. venosa* hemolymph (HLRv/3-100) was via by sodium dodecyl sulfate–polyacrylamide gel electrophoresis (SDS-PAGE) using a 5% stacking gel and a 12% resolving gel, using the Laemmli method with modifications [109]. The samples were dissolved in a Laemmli sample buffer containing 10 mM DTT as a reducing agent. The protein standard marker—a mixture of proteins with molecular weights ranging from 6.5 kDa to 200 kDa (SigmaMarker^TM^, Sigma-Aldrich, Saint Louis, MO, USA)—was used as a molecular marker. For visualization, staining with Coomassie Brilliant Blue G-250 was carried out.

#### 4.2.3. Analyses of the Image of 12% SDS-PAGE Using ImageQuant™ TL v8.2.0 Software

The obtained polyacrylamide gel was captured on an Image Scanner III (GE Healthcare Bio-Sciences AB, Uppsala, Sweden) and the image was opened within the ‘1D gel analysis’ utility of the Image Quant TL v8.2 (GE Healthcare Bio-Sciences AB, Uppsala, Sweden) software, which is a highly automated software for image analysis. All bands were identified manually, including those in the standard protein marker, with a pen tool. To compensate for the intensity of the image background, the background was created through the “image rectangle” setting. The analysis of the molecular weight of each band was performed using data for SigmaMarker^TM^ containing proteins with molecular weights ranging from 6.5 kDa to 200 kDa (SigmaMarker^TM^, Sigma-Aldrich, Saint Louis, MO, USA). Automatically, horizontal bands were matched to the individual bands of the MW marker and calculated with the cubic curve spline. Based on the pre-calculated number of bands in the market, the number of bands that were tested is determined [110].

#### 4.2.4. Matrix-Assisted Laser Desorption/Ionization Time-of-Flight Mass Spectrometry (MALDI-TOF-MS) Analyses of Mucus Fraction (MCa/1-20)

The molecular masses of peptides from the mucus fraction with an MW of 1–20 kDa (MCa/1-20) were analyzed via mass spectrometry (MALDI-TOF MS-analyses) on Autoflex^TM^ III, High-Performance MALDI-TOF, and TOF/TOF Systems (Bruker Daltonics, Bremen, Germany) which uses a 200 Hz frequency-tripled Nd–YAG laser operating at a wavelength of 355 nm. The analysis was carried out using α-cyano-4-hydroxycinnamic acid (CHCA) as a matrix. A total of 2.0 µL of the sample was mixed with 2.0 µL of matrix solution (7 mg/mL of CHCA) in 50% acetonitrile containing 0.1% trifluoroacetic acid, and 1.0 µL of the mixture was spotted on a stainless steel 192-well target plate. The samples were dried at room temperature (25 °C) and subjected to mass analysis. A total of 3500 shots were acquired in the MS mode, and a collision energy of 4200 was applied. The mass spectrometer was externally calibrated with a mixture of angiotensin I (1296.6848 Da), angiotensin II (1046.5418 Da), glufibrinopeptide B (1569.65 Da), ACTH (1–17) B (1569.65 Da), and ACTH (18–39) (2465.1983 Da).

#### 4.2.5. Fungal Strains and Culture Conditions

The fungal strains *A. niger* 17, *P. griseofulvum* 29, and *M. michei*, belonging to the mycological collection of the Stephan Angeloff Institute of Microbiology, Sofia, were used in the experiments. They were selected as Nys- and AmB-resistant strains in our preliminary investigations. The long-term preservation of these fungi was carried out in the Microbank system (Prolab Diagnostics, Richmond Hill, ON, Canada), consisting of sterile vials that contain 25 porous, colored beads and a cryopreservative fluid at −80 °C. Before use, the conidiospores were grown on potato-dextrose agar (PDA) medium at 28 °C for 7 days.

#### 4.2.6. Antifungal Activity Assay

(1)Preparation of Standardized Spore Suspension

Inoculum suspensions were prepared from fresh, mature (5- to 7-day-old) cultures grown on PDA. The resulting spores were added to a sterile solution of Triton X-100 and adjusted to a concentration of 2–6 × 10^5^ spores/mL.

(2)Effect on Spore Development

The antifungal effects of the selected mollusk fractions and commercial drugs against test fungal strains were determined using the broth microdilution method (BMD) with resazurin, an indicator of microbial growth. Typically, 96-well plates were prepared with the addition of 50 μL of potato dextrose broth (PDB) to each well, supplemented with 50 μL of the tested compounds at a final concentration ranging from 0.04 to 16 µg/mL. Then, 10 μL of the spore suspension and 30 μL of 0.02% resazurin were added and plates were incubated for 48 h at 28 °C. The following control samples were used for each variant: (i) negative controls (medium + resazurin without inoculum); (ii) the growth controls (medium + inoculum); positive control (medium + inoculum + test compounds without resazurin). The inhibitory effect was determined through the direct observation of growth and a change in the dye color from blue to pink to purple. Minimum inhibitory concentrations (MICs) were determined as the lowest concentrations that caused complete growth inhibition compared to the control samples without antifungal compounds. All of the experiments were performed in triplicate.

#### 4.2.7. Effect on Fungal Growth Under Submerged Cultivation

For the submerged mycelia production, fungal strains were cultivated in 200 mL Erlenmeyer flasks containing 27 mL PDB and 3 mL spore suspension (2 × 10^8^ spores/mL), with or without the addition of mollusk fractions at a final concentration ranging from 0.01 to 3.0% at 28 °C on a rotary shaker (220 r.p.m.) for 72 h. Samples of biomass content were taken every 24 h. For the oxidative stress biomarkers assay, biomass was harvested in the early stationary growth phase (36 h).

Throughout the cultivation period, mycelia samples were harvested and filtered through a Whatman (Clifton, NJ, USA) No. 4 filter. The separated mycelia were washed twice with distilled water and dried to a constant weight at 105 °C.

#### 4.2.8. Measurement of Cellular Leakage

The twenty-four-hour culture in the early exponential growth phase of *A. niger* and *P. griseofulvum* was washed twice with sterile distilled water to remove residual growth medium and centrifuged at 5000× *g* for 10 min at 4 °C. Subsequently, 1.0 g of the mycelia were exposed to sub-lethal concentrations (0.01, 0.05, 0.1, and 0.3%) of MCa/1-20 or HLRv/3-100. As a negative control, sterile distilled water was used. The protein and reducing sugars released from the tested fungal cells were measured using the Lowry procedure [111] and the Somogy–Nelson method [112], respectively, after incubating at 28 °C for 6 h.

#### 4.2.9. Oxidative Stress Assay

(1)Cell-Free Extract Preparation

The cell-free extract was prepared as described earlier [113]. Briefly, mycelium biomass was harvested via filtration, washed in distilled H_2_O and then in cold 50 mM potassium buffer (pH 7.8), and resuspended in the same buffer. The cell suspension was disrupted by the homogenizer model ULTRA Turax T25 IKA WERK. The temperature during treatment was maintained at 4–6 °C through chilling in an ice-salt bath and filtration through a Whatman filter No. 4 (Clifton, NJ, USA). Cell-free extracts were centrifuged at 13,000× *g* for 20 min at 4 °C.

(2)Measurement of Stress Biomarkers

Protein oxidative damage was measured spectrophotometrically as protein carbonyl content using the 2,4- dinitrophenylhydrazine (DNPH) binding assay [114], slightly modified by Adachi and Ishii [80]. The carbonyl content was calculated using a molar extinction coefficient of 21 mM^−1^ cm^−1^ as nanomoles of DNPH incorporated (protein carbonyls) per mg of protein.

The level of lipid peroxidation was measured using a lipid peroxidation assay kit (Sigma–Aldrich). The lipid degradation caused by oxidative damage was measured by reacting malondialdehyde (MDA) with thiobarbituric acid (TBA) to produce a colorimetric product proportional to the available MDA. The measurement was performed spectrophotometrically at 532 nm. All the experiments were performed in triplicate, with four measurements per sample.

#### 4.2.10. Antioxidant Enzyme Activity Determination

Superoxide dismutase (SOD) activity was measured using the nitro-blue tetrazolium (NBT) reduction method of Beauchamp and Fridovich [115]. Superoxide was measured through the increasing absorbance at 560 nm at 30 °C after 6 min incubation, starting from the beginning of the illumination period. One unit of SOD activity was defined as the amount of enzyme protein required to inhibit the reduction in NBT by 50% (A_560_) and was expressed as units per mg protein [U/mg protein]. Catalase (CAT) activity was determined by monitoring the decomposition of 18 mM H_2_O_2_ at 240 nm [116]. One unit of activity is that which decomposes 1 μmol of H_2_O_2_ min^−1^ mg protein^−1^ at 25 °C and pH 7.0. Specific activity is given as [U/mg protein].

#### 4.2.11. Statistical Evaluation of the Results

The results obtained in this investigation were evaluated through at least three repeated experiments using three parallel runs and the reported values to represent the mean. The error bars indicate the standard deviation (SD) of the mean of triplicate experiments. The data were analyzed using one-way analysis of variance (ANOVA), followed by Tukey’s test. For the statistical processing of the data, the version of the ANOVA software built into the Origin program (OriginPro 2019b, 64-bit) was used.

## 5. Conclusions

In conclusion, the findings of this study reveal, for the first time, the antifungal potential of two fractions of *C. aspersum* and *R. venosa* against Nys- and AmB-resistant strains. The MCa/1-20 and HLRv/3-100 fractions showed a dose-dependent effect on spore and mycelium development during both surface and submerged cultivation. The lethal release of biologically important macromolecules such as proteins and reducing sugars from the treated strains significantly contributed to the suppression and elimination of fungal growth. Additionally, the study highlights the significant role of oxidative stress in the antifungal-damaging mechanism. Increased levels of oxidative stress biomarkers, including lipid peroxidation and oxidatively damaged proteins, were observed, along with a reduction in antioxidant enzyme activity, which corresponded with heightened antifungal efficacy. To our knowledge, this investigation is the first to explore oxidative stress as a factor in the antifungal activity of mollusk-derived fractions.

## Figures and Tables

**Figure 1 ijms-26-00985-f001:**
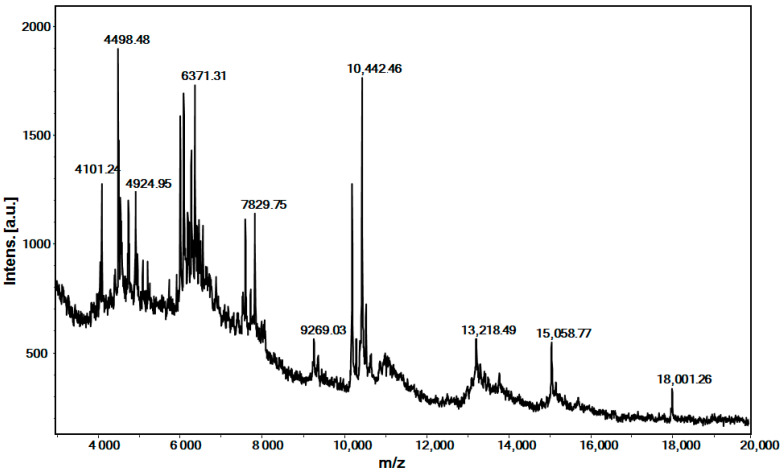
MS spectrum of the fraction with an MW of 1–20 kDa, recorded in the range 3–20 kDa. A standard peptide solution was used to calibrate the mass scale of an Autoflex™ III High-Performance MALDI-TOF and TOF/TOF system (Bruker Daltonics, Bremen, Germany).

**Figure 2 ijms-26-00985-f002:**
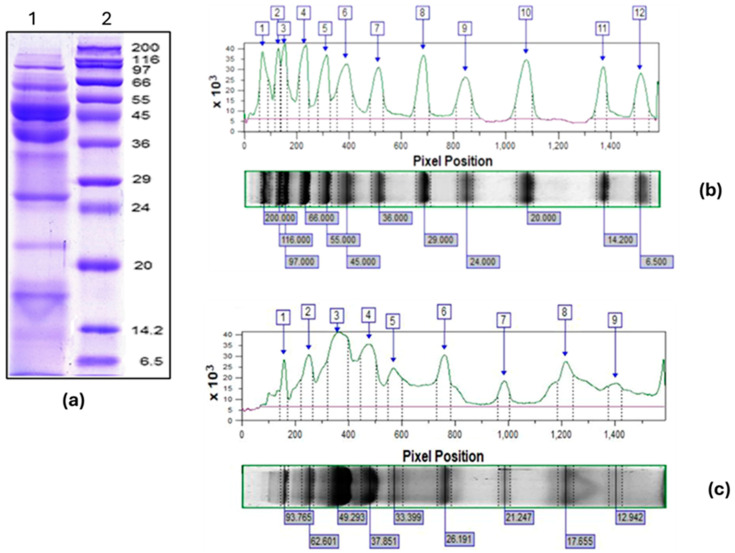
An analysis of HLRv/3-100 on 12% SDS-PAGE, scanned with a high resolution, using ImageQuant^TM^ TL v8.2.0 software. (**a**) Electrophoretic pathway: (1) fraction with an MW of 3–100 kDa from the *R. venosa* hemolymph; (2) standard protein marker with an MW between 6.5 and 200 kDa (SigmaMarkerTM, Sigma-Aldrich, Saint Louis, MO, USA). (**b**) Electrophoretic profile of a standard protein molecular marker (electrophoretic lane 2) analyzed via ImageQuant^TM^ TL. (**c**) Analysis of the electrophoretic profile of the fraction with an MW of 3–100 kDa from the *R. venosa* hemolymph (electrophoretic lane 1) using ImageQuant^TM^ TL.

**Figure 3 ijms-26-00985-f003:**
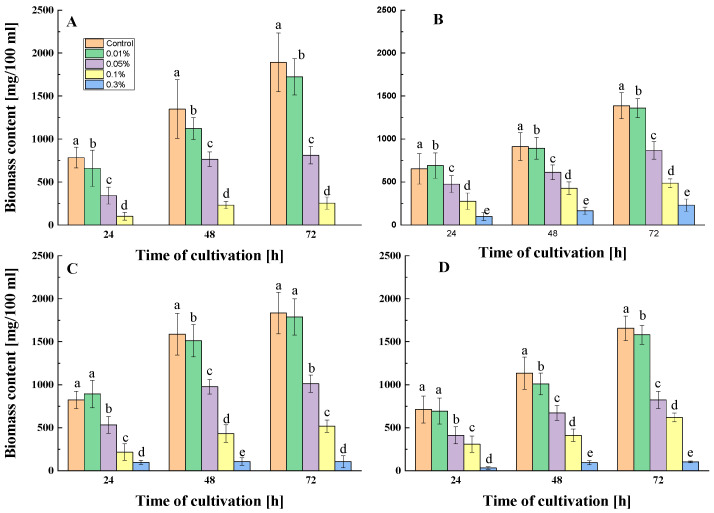
Dose-dependent inhibition effect of MCa/1-20 (**A**,**B**) and HLRv/3-100 (**C**,**D**) against *A. niger* 17 (**A**,**C**) and *P. griseofulvum* 29 (**B**,**D**) under submerged cultivation. Values are the means of three repeated experiments with three replicates in each trial; bars represent the standard deviation. The results indicate a statistically significant reduction in total biomass for all treated variants compared to control (untreated) at every 24 h; a–e represent a significant difference according to Tukey’s test *p* < 0.05.

**Figure 4 ijms-26-00985-f004:**
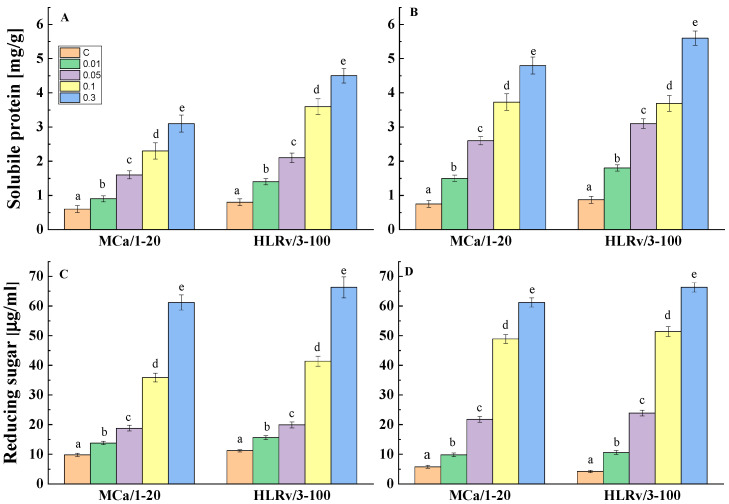
Soluble protein (**A**,**B**) and reducing sugar (**C**,**D**) leakage after treatment of *A. niger* (**A**,**C**) and *P. griseofulvum* (**B**,**D**) with sub-lethal concentrations of MCa/1-20 and HLRv/3-100. Values are means of three repeated experiments with three replicates in each trial; bars represent the standard deviation. Different lower letters (a–e) indicate significant differences (Tukey’s test *p* < 0.05) relative to the control.

**Figure 5 ijms-26-00985-f005:**
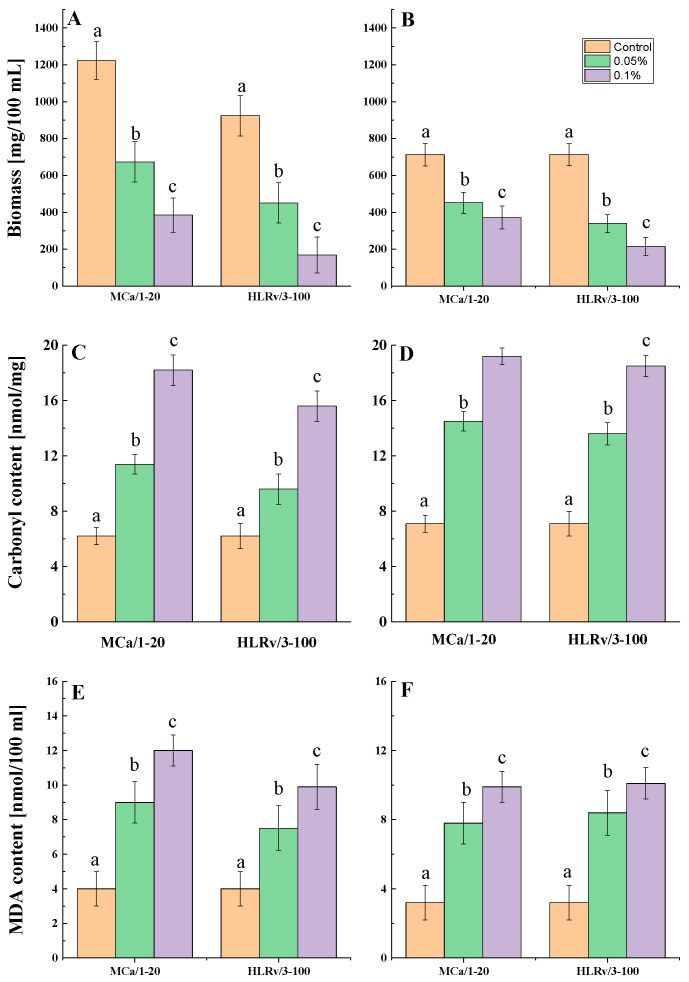
Biomass content (**A**,**B**), oxidatively damaged proteins (**C**,**D**), and MDA content (**E**,**F**) of *A. niger* (**A**,**C**,**D**) and *P. griseofulvum* (**B**,**D**,**F**) afer treatment with sub-lethal concentrations of MCa/1-20 and HLRv/3-100. Values are means of three repeated experiments with three replicates in each trial; bars represent the standard deviation. Different lower letters indicate significant differences (Tukey’s test *p* < 0.05) relative to the control.

**Figure 6 ijms-26-00985-f006:**
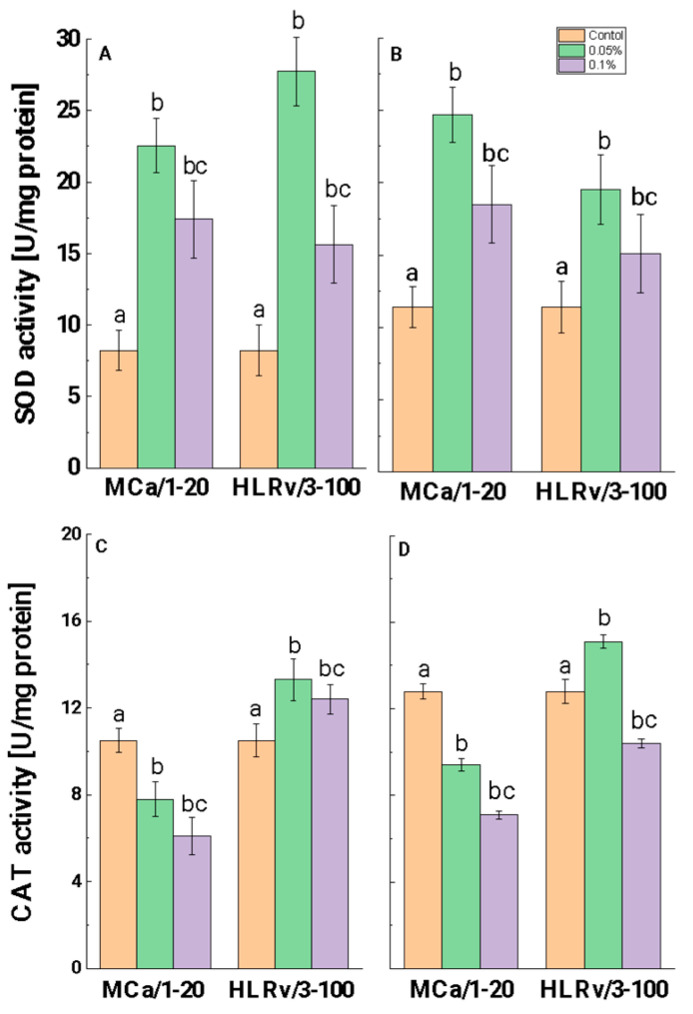
Changes in the activity levels of SOD (**A**,**B**) and CAT (**C**,**D**) of *A. niger* (**A**,**C**) and *P. griseofulvum* (**B**,**D**) after treatment with sub-lethal concentrations of MCa/1-20 and HLRv/3-100. Values are means of three repeated experiments with three replicates in each trial; bars represent the standard deviation. The mollusk fractions had a statistically significant effect on SOD and CAT activity (Tukey, *p* > 0.05). Different lower letters indicate significant differences between treatments versus control and/or lower concentrations.

**Table 1 ijms-26-00985-t001:** The characteristics of peptides with an MW of 1–20 kDa, obtained from a garden snail *C. aspersum*.

No.	Amino Acid Sequence of Peptides	[M + H]^+^ Da	Calcul. Monois. Mass Da	pI	GRAVY	Net Charge	Predicted by iAMPpred Software		
							Antibacterial (%)	Antiviral (%)	Antifungal (%)
1 ^a^	LLPFKEPDL	1071.61	1070.60	4.37	−0.600	−2/+1	28	51	19
2 ^a^	ACGATLQLENCG	1179.78	1178.51	4.00	+0.350	-1/0	29.3	35.7	43.7
3 ^a^	LNLGGNGANGLVGG	1212.76	1211.63	5.52	+0.321	0/0	74.0	43.1	74.3
4 ^a^	AGVGGAAGNPSTYVG	1277.71	1276.60	5.57	+0.260	0/0	25.1	7.4	11.3
5 ^a^	GGGMVKEDGSCLGV	1308.77	1307.58	4.37	+0.207	−2/+1	40.4	31.4	33.7
6 ^b^	MLGGGVNSLRPPK	1325.80	1324.73	11.0	−0.262	0/+2	22.8	14.0	8.9
7 ^a^	CVGGAGGHGDSCAKGT	1376.53	1375.56	6.73	−0.106	−1/+1	85.2	48.8	74.4
8 ^a^	GGGGYHTWGEGGKF	1409.48	1408.62	6.75	−0.964	−1/+1	69.0	62.8	72.6
9 ^a^	MLNVAVNKGEVKH	1438.87	1437.78	8.37	−0.138	−1/+2	56.4	38.0	19.7
10 ^c^	NLVGGSGGGGRGGANPLG	1496.79	1495.75	9.75	−0.217	0/+1	66.0	33.7	48.2
11 ^a^	GTMSPAGGEMGPVTAGVG	1576.04	1574.71	4.00	+0.250	−1/0	13.1	24.8	8.3
12 ^a^	GTKGCGPGSCPPGDTVAGVG	1716.79	1715.76	5.82	−0.100	−1/+1	23.8	20.2	25.8
13 ^d^	ACSLLLGGGGVGGGKGGGGHAG	1738.97	1737.86	8.27	+0.409	0/+1	82.6	49.5	67.0
14 ^b^	LLLDGFGGGLLVEHDPGS	1796.00	1794.92	4.02	+0.439	-3/0	37	45	10
15 ^e^	MGGWGGLGGGHNGGWMPPK	1852.97	1851.83	8.52	−0.611	0/+1	69	56.0	57.0
16 ^e^	ACLTPVDHFFAGMPCGGGP	1876.88	1875.81	5.08	+0.542	−1/0	32.4	42.6	20.1
17 ^e^	NGLFGGLGGGGHGGGGKGPGEGGG	1909.99	1908.88	6.75	−0.487	−1/+1	89.5	67.2	79.5
18 ^a^	LLLDNKGGGLVGGLLGGGGKGGG	1966.11	1965.10	8.59	+0.322	−1/+2	93	56	81
19 ^a^	GMVLLHCSPALDFHKTPAV	2036.09	2035.04	6.91	+0.616	−1/+1	18	53	14
20 ^a^	LPFLLGVGGLLGGSVGGGGGGGGAPL	2136.24	2135.17	5.52	+1.023	0/0	66	33	36
21 ^a^	MVLDGKGGGGLLGGVLGGGKDAHLGG	2292.33	2291.21	6.50	+0.319	−2/+2	84.3	59.0	71
22 ^a^	LLKDNGVGGLLGGGGAGGGGLVGGNLGGGAG	2478.40	2477.30	5.84	+0.439	−1/+1	86.4	54	66
23 ^a^	KTSKLMVYLAGGGGGLLGGVGGGGGGAGGGGPGGL	2843.76	2842.48	9.70	+0.374	0/+2	76	47	67

^a^ The AASs of these peptides were also identified previously by Velkova et al. [47]; ^b^ the AASs of these peptides were identified previously by Dolashki et al. [48]; ^c^ the AASs of these peptides were also identified previously by Dolashki et al. [49]; ^d^ the AASs of these peptides were identified previously by Topalova et al. [50]; ^e^ the AASs of these peptides were identified previously by Velkova et al. [51].

**Table 2 ijms-26-00985-t002:** MIC values for MCa/1-20 and HLRv/3-100 against *A. niger* 17, *P. griseofulvum* 29, and *M. michei*.

Strain	Test Compound/MIC [μg/mL]			
	AmB	Nys	HLRv/3-100	MCa/1-20
*A. niger*	7	4	3.50	1.75
*P. griseofulvum*	4	2	1.75	1.75
*M. michei*	8	2	3.50	3.50

Ns and AmB were used as positive controls.

## Data Availability

Data are contained within the article.

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
