# Peer review of "The Role of Oxidative Stress in the Antifungal Activity of Two Mollusk Fractions on Resistant Fungal Strains"

_ijms, 2025, doi:10.3390/ijms26030985_

Round 1
Reviewer 1 Report
Comments and Suggestions for Authors
Manuscript Number: IJBIOMAC-D-24-18352
Title: The Role of Oxidative Stress in Antifungal Activity of Two Mollusc Fractions on Resistant Fungal Strains
The aim of this study was to evaluate the antifungal activity of novel mollusc fractions against resistant fungal strains and to investigate the role of oxidative stress in their mechanism of action. Specific fractions of garden snail Cornu aspersum and marine snail Rapana venosa were characterized and found to have significant fungicidal effects against Penicillium griseofulvum and Aspergillus niger. Compared to the control group, the treated group showed an increase in intracellular release of proteins and reducing sugars, along with an increase in the levels of oxidative stress biomarkers and a down-regulation of antioxidant enzyme defenses. This specifies that the extracted proteins exhibited strong antifungal activity.
However, this research article is notable for the improvements that the English language could be improved to more clearly express the research. Therefore, the manuscript needs to major revise for possible publish in International Journal of Biological Macromolecules.
1. In line 21, why is “SDS-PAGE and” bolded?
2. References need to be inserted in lines 32-40.
3. Line 33, change “World Health Organization“ to ”World Health Organization (WHO)“。
4. Line 121, “A. niger 17, P. griseofulvum 29 and M. michei” and line 127, “C. aspersum” are first appearances and need to be described with their full names. Also, this problem exists elsewhere in the manuscript, please double-check.
5. Lines 247-248, change“Complete inhibition was achieved at 0.5% “ to ”Complete inhibition was achieved at 0.5%.“。
6. The words for the vertical coordinates of Figures 4A and B are incorrect, please correct them.
7. In line 268, “Comparable” should not be italicized.
8. Why did A. niger and P. griseofulvum exhibit lower SOD and CAT contents after treatment with 0.1% concentrations of MCa/1-20 and HLRv/3-100 compared to 0.05%?
9. It is recommended that Figures 4, 5, 6 and 7 be merged, also Figures 8 and 9. Because of their depiction of the relevant content.
10. The language of the article is redundant and often confusing at times, such as in the introduction section where the author is always emphasizing the issue of resistance over and over again, which can cause reading fatigue for the person reading it. The same problem exists in the presentation of the experimental results, hopefully the author will be able to condense the manuscript.
11. Lines 609-615, the line spacing is incorrect.
12. When examining the effect on spore development and the effect on fungal growth, incubation at 28°C for 45-72 h is required. The protein may be degraded under these conditions, which could affect the stringency of the experiments. It is recommended to verify whether this may be the case.
Comments on the Quality of English LanguageEnglish language could be improved.
Author Response
Dear Reviewer,
Thank you very much for taking the time to review this manuscript! Please find the detailed responses below and the corresponding revisions highlighted in red in the re-submitted file.
General comments. The aim of this study was to evaluate the antifungal activity of novel mollusc fractions against resistant fungal strains and to investigate the role of oxidative stress in their mechanism of action. Specific fractions of garden snail Cornu aspersum and marine snail Rapana venosa were characterized and found to have significant fungicidal effects against Penicillium griseofulvum and Aspergillus niger. Compared to the control group, the treated group showed an increase in intracellular release of proteins and reducing sugars, along with an increase in the levels of oxidative stress biomarkers and a down-regulation of antioxidant enzyme defences. This specifies that the extracted proteins exhibited strong antifungal activity.
However, this research article is notable for the improvements that the English language could be improved to more clearly express the research. Therefore, the manuscript needs to major revise for possible publish in International Journal of Biological Macromolecules.
Response 1. We agree with the reviewer’s assessment. Thank you for pointing this out. The manuscript has been substantially revised following the reviewers' comments. Notably, the Introduction, Results, and Discussion sections have been extensively updated. Additionally, comprehensive corrections have been made throughout the manuscript, with particular attention given to the quality of the English language.
Comments 1. In line 21, why is “SDS-PAGE and” bolded?
Response 1. This error arose during the process of transferring the file between computers. The error has been corrected.
Comments 2. References need to be inserted in lines 32-40.
Response 2. Agree. This passage has been revised; references have been included.
Comments 3. Line 33, change “World Health Organization“ to ”World Health Organization WHO“
Response 3. Change has been made.
Comments 4. Line 121, “A. niger 17, P. griseofulvum 29 and M. michei” and line 127, “C. aspersum” are first appearances and need to be described with their full names. Also, this problem exists elsewhere in the manuscript, please double-check.
Response 4. We agree. All Latin names have been corrected according to their first appearance in the text.
Comments 5. Lines 247-248, change “Complete inhibition was achieved at 0.5% “ to” Complete inhibition was achieved at 0.5%.“
Response 5. The correction has been made (i.e. point has been added).
Comments 6. The words for the vertical coordinates of Figures 4A and B are incorrect, please correct them.
Response 6. The correction has been made.
Comments 7. In line 268, “Comparable” should not be italicized.
Response 7. The correction has been made.
Comments 8. Why did A. niger and P. griseofulvum exhibit lower SOD and CAT contents after treatment with 0.1% concentrations of MCa/1-20 and HLRv/3-100 compared to 0.05%?
Response 8. You have raised an important point here. Several studies have shown that higher concentrations of antifungals (fungicidal doses) inhibit both antioxidant enzymes, SOD and CAT. Similar results have been reported for a wide range of natural extracts. Probably, excessive activation of stress responses is unable to protect the cells but rather can cause cellular death by depleting ATP and generating ROS [Kobayashi et al., 2002]. Moreover, the decrease in CAT activity can be ascribed to the cytotoxic impacts of the accumulating ROS, which made a substantial contribution to this phenomenon [Reymick et al., 2024]. Antifungal fractions may also suppress transcriptional regulators or act directly on genes to prevent biosynthesis and/or metabolism [Reymick et al., 2024].
It is important to note that certain antimicrobials, including antifungal agents, exhibit both antioxidant and prooxidant characteristics, which are influenced by concentration, cell type, duration of exposure, and environmental conditions [Valenzuela-Cota et al., 2019]. Research has shown that prooxidant effects are particularly evident at elevated concentrations (Rajashekar, 2023). Moreover, prooxidant activity is more commonly displayed by compounds with smaller molecules than by those with larger molecules (Hagerman et al., 1998; Pesarico et al., 2013; Blokhina et al., 2003).
The above information has been used in the section “Discussion”.
References:
Kobayashi, D.; Kondo, K.; Uehara, N.; Otokozawa, S.; Tsuji, N.; Yagihashi, A.; Watanabe, N. Endogenous Reactive Oxygen Species is an Important Mediator of Miconazole Antifungal Effect. Antimicrob. Agents Chemother. 2002, 46, 3113-3117. doi: 10.1128/AAC.46.10.3113-3117.2002
Reymick, O.O.; Liu, D.; Tan, X.; OuYang, Q.; Tao, N. Cuminaldehyde Downregulates Folate Metabolism and Membrane Proteins to Inhibit Growth of Penicillium digitatum in Citrus Fruit. Future Postharv. Food 2024, 1, 104-123. DOI: 10.1002/fpf2.12010
Valenzuela-Cota, D.F.; Buitimea-Cantúa, G.V.; Plascencia-Jatomea, M.; Cinco-Moroyoqui, F.J.; Martínez-Higuera, A.A.; Rosas-Burgos, E.C. Inhibition of the Antioxidant Activity of Catalase and Superoxide Dismutase from Fusarium verticillioides Exposed to a Jacquinia macrocarpa Antifungal Fraction. J. Environmen. Sci. Health, Part B 2019, 54, 647-654. doi:10.1080/03601234.2019.1622978
Rajashekar, C.B. Dual Role of Plant Phenolic Compounds as Antioxidants and Prooxidants. Am. J. Plant Sci. 2023, 14, 15-28. https://www.scirp.org/journal/ajps
Hagerman, A.E.; Riedi, K.M.; Jones, G.A.; Sovik, K.N.; Ritchard, N.T.; Hartzfeld, P.W.; Riechel, T.L. High Molecular Weight Polyphenolics Tannins as Biological Antioxidants. J. Agricult. Food Chem. 1998, 46, 1887-1892. https://doi.org/10.1021/jf970975b
Blokhina, O.; Virolainen, E.; Fagerstedt, K.V. Antioxidants, Oxidative Damage and Oxygen Deprivation Stress: A Review. Ann. Bot. 2003, 91, 179-194. https://doi.org/10.1093/aob/mcf118
Comments 9. It is recommended that Figures 4, 5, 6 and 7 be merged, also Figures 8 and 9. Because of their depiction of the relevant content.
Response 9. Figures 5, 6, and 7 have been merged into Figure 5; Figures 8 and 9 have been merged into Figure 6.
Comments 10. The language of the article is redundant and often confusing at times, such as in the introduction section where the author is always emphasizing the issue of resistance over and over again, which can cause reading fatigue for the person reading it. The same problem exists in the presentation of the experimental results, hopefully, the author will be able to condense the manuscript.
Response 10. We agree with the reviewer. A large part of the sections Introduction, Results, and Discussion has been rewritten according to the comments of both reviewers. The entire manuscript was edited in English.
Comments 11. Lines 609-615, the line spacing is incorrect.
Response 11. Thank you for the comments. The correction has been made.
Comments 12. When examining the effect on spore development and the effect on fungal growth, incubation at 28°C for 45-72 h is required. The protein may be degraded under these conditions, which could affect the stringency of the experiments. It is recommended to verify whether this may be the case.
Response 12. You have highlighted a significant question, thank you. The evaluation of protein leakage from fungal cells treated with mollusc fractions was performed using 24-hour cultures. Subsequently, 1.0 g of mycelia was subjected to sub-lethal concentrations of the two fractions being tested. The release of protein and reducing sugars from the fungal cells was quantified after incubation at 28°C for both 6 and 24 hours. The results obtained from the 6-hour and 24-hour treatments were comparable; therefore, only the data from the 6-hour incubation is presented in this manuscript. Comparable experimental protocols have been reported in the literature.
Our earlier investigations into oxidative stress induction within fungal cells indicate that using mycelia during the early exponential phase is most effective for assessing modifications in intracellular components. Moreover, a duration of 6 hours of exposure is sufficient to achieve significant changes.
Comments 13. Comments on the Quality of English Language: The English language could be improved.
Response 13. Thank you for the comments. Substantial corrections have been made to the Introduction, Results, and Discussion sections, reflecting the suggestions from both reviewers. Furthermore, the manuscript has been comprehensively edited for the English language.
Reviewer 2 Report
Comments and Suggestions for Authors
Authors investigate the potential of mollusk fractions as new antifungal drugs and examine the presence of oxidative stress in fugi treated with these fractions. I have some suggestions and concerns as outlined below:
Line 40-43: Text a bit convoluted and ambiguous. Is MDR only against chemotherapeutic drugs? Because the text also mentions cancer cells it becomes unclear if in the case of infections, the host cells or the parasite cells that express the proteins.
Line 46: Only about 150 thousand fungi species have been identified. Estimates could go from 2 to over 10 million. Please add citations and be more precise on your writing.
Line 76: why is “Similarly, peptides isolated from” italicised?
Line 105: Is the antifungal action of all the different compounds solely caused by oxidative stress? Or is oxidative stress part of a broader mechanism of action? Or is oxidative stress a consequence of the mechanism of action but has no causative role?
Fig3, Fig 4, Fig 5, Fig 6, Fig 7, Fig 8, Fig 9: include information about the statistical test used (ANOVA?). There is no indication of which groups are statistically different from which in these figures. Please, also adjust the language to the use of more appropriate scientific terms – legends often finish with a sentence that reads “Mollusk fractions turn out to have a statistically significant effect on…”
My two main concerns are regarding the mechanism of action and the safety of the fractions for human use:
Authors have clearly demonstrated the presence of oxidative stress. However, it was not demonstrated that oxidative stress has a causative role. The target of the mollusk peptides was not established, so there is no base to assume the mechanism of action is centred on oxidative stress. A vast number of mechanisms that cause cell toxicity will lead to ROS generation as a byproduct but eliminating ROS in these scenarios do not prevent cell death, meaning ROS is not the cause. The authors spend a great deal of time talking about how other antifungals cause oxidative stress, but that is no proof of concept either. To demonstrate that oxidative stress is causative it has to be eliminated/reduced (by overexpressing antioxidant enzymes, using antioxidant treatment, etc) and then showed that fungi survive treatment with the mollusk fractions. Oxidative stress also has multiple origins: mitochondria; Nox/Duox; peroxisomes; NOS; P450s; endoplasmic reticulum, etc. But the authors do not investigate that. Altogether, there is no evidence to claim that the mechanism of damage has been determined. The only well-established claim is that oxidative stress is present, but whether it is the sole cause, part of the cause, or a consequence of the mechanism of damage remains unknown.
The authors discuss the importance of identifying new molecules/drugs to curb fungal resistance. But there is no mention to what effect these mollusk fraction could have in humans. Given the cellular targets are unknown and oxidative stress is generalised, one wonders how safe these fractions would be for human use. This point and the previous one must be carefully considered and the manuscript adjusted accordingly.
Comments on the Quality of English LanguageText is sometimes convoluted and sentences too long. That makes the text at a times not easy to follow. Please see suggestions where I point a few examples that could be improved.
Author Response
Dear Reviewer,
Thank you very much for taking the time to review this manuscript! Please find the detailed responses below and the corresponding revisions highlighted in red in the re-submitted file.
General comments. Authors investigate the potential of mollusk fractions as new antifungal drugs and examine the presence of oxidative stress in fungi treated with these fractions. I have some suggestions and concerns as outlined below:
Comments 1. Line 40-43: Text a bit convoluted and ambiguous. Is MDR only against chemotherapeutic drugs? Because the text also mentions cancer cells it becomes unclear if in the case of infections, the host cells or the parasite cells that express the proteins.
Response 1: The passage has been revised to give additional information on the subject of drug resistance. The following text has been added (pages 2-3, lines 43-52):
AMR occurs when microorganisms adapt in response to the use of drugs designed to inhibit or kill them [2]. Due to drug resistance, antimicrobial agents lose their efficacy, rendering infections progressively challenging or unmanageable to treat. The development of simultaneous multidrug resistance (MDR) is characterized by the acquired non-susceptibility to at least one agent across three or more antimicrobial groups [3,4]. Microbes exhibit several resistance mechanisms, including inherent resistance to specific antimicrobials, genetic mutations, and acquired resistance from other species [4]. The presence of resistant microorganisms complicates treatment strategies, often requiring the use of alternative or higher doses of antimicrobials or leading to a shortage of effective treatment options.
Comments 2: Line 46: Only about 150 thousand fungi species have been identified. Estimates could go from 2 to over 10 million. Please add citations and be more precise in your writing.
Response 2: Agree. We have, accordingly, modified the passage of the Introduction (page 3, lines 48-54) to emphasize this point. The following text has been added:
Fungi are ubiquitous and occupy all possible ecological niches. Their number is not exactly known. The most reported data for fungal species are in the range of 1.5 to 10 million depending on the method of calculation (Cazabonne et al., 2024). Based on comparative analyses with plants, the number of fungal species ranges between 2.2 and 3.8 million (Hawksworth and Lücking, 2017; Wijayawardene et al., 2024). However, high-throughput DNA sequencing suggests a significantly higher number, between 11.7 and 13.2 million (Wu et al., 2019; Hyde, 2022).
References:
Cazabonne, J.; Walker, A.K.; Lesven, J.; Haelewaters, D. Singleton-based species names and fungal rarity: Does the number really matter? IMA Fungus 2024, 15, 7. 7 https://doi.org/10.1186/s43008-023-00137-2
Hawksworth, D.L.; Lücking, R. Fungal Diversity Revisited: 2.2 to 3.8 Million Species. Microbiol. Spectr. 2017, 5, FUNK-00522016. doi: 10.1128/microbiolspec.FUNK-0052-2016
Hyde, K.D. The numbers of fungi. Fungal Diversity 2022, 114, 1. https://doi.org/10.1007/s13225-022-00507-y
Wijayawardene, N.N.; Hyde K.D.; Mikhailov, K.V.; Péter, G.; Aptroot, A.; Pires‑Zottarell, C.L.A.; Goto,·B.T.; Tokarev, Y.S.; et al. Classes and phyla of the kingdom Fungi. Fungal Diversity 2024, 127, 1-165. https://doi.org/10.1007/s13225-024-00540-z
Wu, B.; Hussain, M.; Zhang, W.; Stadler, M.; Liu, X.; Xiang, M. Current insights into fungal species diversity and perspective on naming the environmental DNA sequences of fungi. Mycology 2019, 10, 127-140. doi: 10.1080/21501203.2019.1614106.
Comments 3: Line 76: why is “Similarly, peptides isolated from” italicised?
Response 3: The font has been corrected.
Comments 4: Line 105: Is the antifungal action of all the different compounds solely caused by oxidative stress? Or is oxidative stress part of a broader mechanism of action? Or is oxidative stress a consequence of the mechanism of action but has no causative role?
Response 4: Thank you for your very important question. The published data does not provide a definitive answer to this question. Several reports underlined different mechanisms of the antifungal effect of natural products or chemical drugs. For instance, these agents may inflict damage on mitochondria, efflux pumps, and membranes, in addition to inhibiting DNA and RNA, protein synthesis, and ergosterol synthesis (Silva-Beltrán et al., 2023; Zhang et al., 2023; Long and Li, 2024
Regarding the function of ROS, several studies have demonstrated that they play a key role in fungicidal activities and may be involved in drug-induced programmed cell death (PCD). However, it remains ambiguous whether the ROS are a direct consequence of the antifungal mechanism of action or a secondary consequence of drug-induced PCD (Gonzalez-Jimenez et al., 2023). According to Da et al. (2019), the antifungal activity of the extract from Scutellaria baicalensis Georgi root can be attributed to the accumulation of intracellular ROS. Similar suggestions have been made by Silva-Beltrán et al. (2023) regarding the antifungal mechanism of action of a range of natural products, including plant extracts, propolis, phenolic compounds, alkaloids, essential oils, etc. The role of ROS has been further corroborated by several studies, which indicate their involvement in the fungicidal activities of drugs and their potential role in drug-induced PCD. The generation of ROS is considered one of the main biochemical reasons for apoptosis (Da et al., 2019; Gonzalez-Jimenez et al., 2023). The hypothesis that oxidative stress is the primary mechanism underlying antifungal activity has been supported by a number of authors in the field (Khan et al., 2011; de Araújo Neto et al., 2020; Wu et al., 2023; Liu et al., 2024).
Hawever, there is a notable scarcity of information concerning the action mechanisms of antimicrobial peptides (AMPs) derived from molluscs.
The above information has been used in the section “Discussion”.
References
Da, X.; Nishiyama, Y.; Tie, D. et al. Antifungal activity and mechanism of action of Ou-gon (Scutellaria root extract) components against pathogenic fungi. Sci. Rep. 2019, 9, 1683. https://doi.org/10.1038/s41598-019-38916-w
de Araújo Neto L.N.; do Carmo, M.; de Lima, A.; de Oliveira, J.F.; de Souza, E.R.; Machado, S.E.F. et al. Thiophene-thiosemicarbazone derivative (L10) exerts antifungal activity mediated by oxidative stress and apoptosis in C. albicans. Chem. Biol. Interact. 2020, 320, 109028. https://doi.org/10.1016/j.cbi.2020.109028
Gonzalez-Jimenez, I.; Perlin, D.S.; Shor, E. Reactive oxidant species induced by antifungal drugs: identity, origins, functions, and connection to stress-induced cell death. Front. Cell. Infect. Microbiol. 2023, 13, 1276406. doi: 10.3389/fcimb.2023.1276406. PMID: 37900311; PMCID: PMC10602735
Khan, A.; Ahmad, A.; Akhtar, F.; Yousuf, S.; Xess, I.; Khan, L.A.; Manzoor, N. Induction of Oxidative Stress as a Possible Mechanism of the Antifungal Action of Three Phenylpropanoids. FEMS Yeast Res. 2011, 11, 114–122. DOI:10.1111/j.1567-1364.2010.00697.x
Liu, J.; Wang, L.; Sun, Y.; Xiong, Y.; Li, R.; Sui, M.; Gao, Z.; Wang, W.; Sun, H.; Dai, J. Antifungal Activity and Multi-Target Mechanism of Action of Methylaervine on Candida albicans. Molecules 2024, 29, 4303. https://doi.org/10.3390/ molecules29184303
Long, N.; Li, F. Antifungal Mechanism of Natural Products Derived from Plants: A Review. Nat. Prod. Communic. 2024; 19. doi:10.1177/1934578X241271747
Silva-Beltrán N. P., Boon, S.A.; Ijaz, M.K.; McKinney, J.; Gerba, C.P. Antifungal activity and mechanism of action of natural product derivates as potential environmental disinfectants. J. Ind. Microbiol. Biotechnol. 2023, 50, kuad036. https://doi.org/10.1093/jimb/kuad036
Wu, Y.; Chen, Y.; Lu, H.; Ying, C. Miltefosine exhibits fungicidal activity through oxidative stress generation and Aif1 activation in Candida albicans. Int. J. Antimicrob. Agents 2023, 62, 106819. https://doi.org/10.1016/j.ijantimicag.2023.106819
Comments 5: Fig 3, Fig 4, Fig 5, Fig 6, Fig 7, Fig 8, Fig 9: include information about the statistical test used (ANOVA?). There is no indication of which groups are statistically different from which in these figures. Please, also adjust the language to the use of more appropriate scientific terms – legends often finish with a sentence that reads “Mollusk fractions turn out to have a statistically significant effect on…”
Response 5: In response to Reviewer 1, Figures 5, 6, and 7 are combined into Figure 5; and Figures 8 and 9 are combined into Figure 6. The legends of Figures 3, 4, 5, and 6 have been improved following the reviewer's comments.
Comments 6: There are also typos, and the text should be check for that, for instance line 147: "These results are agreement" is missing "in".
Response 6: The correction has been done
Comments 7: Authors have clearly demonstrated the presence of oxidative stress. However, it was not demonstrated that oxidative stress has a causative role. The target of the mollusk peptides was not established, so there is no base to assume the mechanism of action is centred on oxidative stress. A vast number of mechanisms that cause cell toxicity will lead to ROS generation as a byproduct but eliminating ROS in these scenarios do not prevent cell death, meaning ROS is not the cause. The authors spend a great deal of time talking about how other antifungals cause oxidative stress, but that is no proof of concept either. To demonstrate that oxidative stress is causative it has to be eliminated/reduced (by overexpressing antioxidant enzymes, using antioxidant treatment, etc) and then showed that fungi survive treatment with the mollusk fractions. Oxidative stress also has multiple origins: mitochondria; Nox/Duox; peroxisomes; NOS; P450s; endoplasmic reticulum, etc. But the authors do not investigate that. Altogether, there is no evidence to claim that the mechanism of damage has been determined. The only well-established claim is that oxidative stress is present, but whether it is the sole cause, part of the cause, or a consequence of the mechanism of damage remains unknown.
Response 7: Thank you for your reasonable comment. We agree that the oxidative stress could be provoked by various endo- and exogenic sources. As is known, endogenic sources include mitochondria, peroxisomes, P450s; endoplasmic reticulum, etc. To date, evidence for the involvement of peroxisomes, ERs, and NADPH oxidases in the generation of ROS induced by antifungal agents is extremely limited (Gonzalez-Jimenez et al., 2023). Several papers have suggested that ROS generated after fungicide treatment are of mitochondrial origin (Shekhova et al., 2017; Haque et al., 2019, Muñoz-Megías et al., 2023]. Zhu et al (2023) demonstrated that inhibition of mitochondrial function using specific inhibitors or genetic manipulation led to a decrease in ROS levels following treatment with amphotericin B and itraconazole. Similar experiments with MCa/1-20 and HLRv/3-100 could be the subject of further study.
In the present manuscript, we focused on the possibility of mollusc fractions inducing oxidative stress. The basis for our assumption that mollusc AMPs induce oxidative stress in A. niger and P. griseofulvum gives us a comparison with control variants that were placed under the same conditions as the test variants but without being treated with antifungal fractions. The significant difference in the results between the two groups on oxidative stress biomarkers and antioxidant enzyme defence activity is likely due to the presence and absence of the antifungal agent. In addition, the increased concentrations of MCa/1-20 and HLRv/3-100 lead to a serious increase in the level of lipid peroxidation and oxidatively damaged proteins, which is an indication of the accelerated generation of ROS.
The above information has been used in the section “Discussion”.
References:
Gonzalez-Jimenez, I.; Perlin, D.S.; Shor, E. Reactive oxidant species induced by antifungal drugs: identity, origins, functions, and connection to stress-induced cell death. Front. Cell. Infect. Microbiol. 2023, 13, 1276406. doi: 10.3389/fcimb.2023.1276406. PMID: 37900311; PMCID: PMC10602735
Haque, F.; Verma, N.K.; Alfatah, M.; Bijlanik, S.; Bhattacharyya, M.S. Sophorolipid exhibits antifungal activity by ROS mediated endoplasmic reticulum stress and mitochondrial dysfunction pathways in Candida albicans. RSC Adv. 2019, 9, 41639–41648. DOI: 10.1039/c9ra07599b
Muñoz-Megías, M. L.; Sánchez-Fresneda, R.; Solano, F.; Maicas, S.; Martínez-Esparza, M.; Argüelles, J.C. The antifungal effect induced by itraconazole in Candida parapsilosis largely depends on the oxidative stress generated at the mitochondria. Curr. Genet. 2023, 69, 165–173. doi: 10.1007/s00294-023-01269-z
Shekhova, E.; Kniemeyer, O.’; Brakhage, A.A. Induction of Mitochondrial Reactive Oxygen Species Production by Itraconazole, Terbinafine, and Amphotericin B as a Mode of Action against Aspergillus fumigatus. Antimicrob Agents Chemother. 2017, 61, e00978-17. doi: 10.1128/AAC.00978-17.
Zhu, G.; Chen, S.; Zhang, Y.; Lu, L. Mitochondrial Membrane-Associated Protein Mba1 Confers Antifungal Resistance by Affecting the Production of Reactive Oxygen Species in Aspergillus fumigatus. Antimicrob. Agents Chemother. 2023, 67, e00225-23. https://doi.org/10.1128/aac.00225-23
Comments 8: The authors discuss the importance of identifying new molecules/drugs to curb fungal resistance. But there is no mention to what effect these mollusk fraction could have in humans. Given the cellular targets are unknown and oxidative stress is generalised, one wonders how safe these fractions would be for human use. This point and the previous one must be carefully considered and the manuscript adjusted accordingly.
Response 8: Thank you for the very important question“ how safe these fractions would be for human use?”. In recent years, a number of studies have commented on the safety of snail mucus (mainly A. fulica and H. aspersa Müller, also known as C. aspersum) as an antimicrobial agent, wound healing agent, an active ingredient in cosmetic and pharmaceutical products [Aouji et al., 2023; Herman et al., 2024; Singh et al., 2024].
Several studies have highlighted the absence of cytotoxic effects in different cell lines such as human keratinocytes, human dermal fibroblasts (MRC-5), and murine embryo fibroblasts (NIH-3T3) [Gentili et al., 2020; Trapella et al., 2019]. The study by Trapella et al. demonstrated the lack of any cytostatic effect on fibroblasts treated in vitro by mucus extract (from H. aspersa) with concentrations in the range of 4–400 µg/mL. It is obvious that the concentrations used in our study for both fractions (0.04 to 16 µg/mL) are much lower than the highest safe concentration.
The lack of cytotoxic effect of mucus extract of H. aspersa in humans has led to the creation of a new lubricating ophthalmic solution eye GlicoPro in alleviating severe dry eye disease [Singh et al., 2024; Ballesteros-Sánchez et al., 2024]. Results by Mencucci et al. (2021) showed that mucus extract used in GlicoPro was characterized by the presence of a complex mixture of different biological molecules such as high and low molecular weight proteins, glycosaminoglycans, and secondary metabolites, including peptides [Mencucci et al., 2021].
Our results also prove the lack of cytotoxicity of the mucus fraction on human skin fibroblasts (BJ) and human keratinocytes (HaCaT), which are shown in our new study "Synergistic antibacterial effect of the mucus fraction of C. aspersum with antibiotics against pathogenic bacteria isolated from wounds of diabetic patients", submitted to Antibiotics (MDPI).
Considering the composition of the fraction HLRv/3-100 (included in the manuscript) and the used concentrations in a range of 0.04 - 16 µg/mL against the fungal pathogens, we assume that the fraction could not have dangerous cytotoxicity in humans, but this will be the subject of further studies - evaluation of the cytotoxicity of HLRv/3-100 on 2 human cell lines - BJ and HaCaT.
The use of mucus fraction MCa/1-20 or hemolymph fraction HLRv/3-100 in an antifungal formulation requires long-term studies, including additional in vitro and in vivo studies.
References:
Aouji, M.; Rkhaila, A.; Bouhaddioui, B.; Zirari, M.; Harifi, H.; Taboz, Y.; Lrhorfi, LA.; Bengueddour, R. Chemical composition, mineral profile, anti-bacterial, and wound healing properties of snail slime of Helix aspersa Müller. Biomedicine (Taipei). 2023, 13(4), 10-19. doi: 10.37796/2211-8039.1424.
Herman, A.; Wińska, P; Białek, M.; Herman, A.P. Biological Properties of the Mucus and Eggs of Helix aspersa Müller as a Potential Cosmetic and Pharmaceutical Raw Material: A Preliminary Study. Int J Mol Sci. 2024, 25(18), 9958. https://doi.org/10.3390/ijms25189958
Singh, N.; Brown, AN.; Gold, M.H. Snail extract for skin: A review of uses, projections, and limitations. J Cosmet Dermatol. 2024, 23(4), 1113-1121. https://doi.org/10.1111/jocd.16269
Gentili, V.; Bortolotti, D.; Benedusi, M.; Alogna, A.; Fantinati, A.; Guiotto, A.; Turrin, G.; Cervellati, C..; Trapella, C.; Rizzo, R.; Valacchi, G. HelixComplex snail mucus as a potential technology against O3 induced skin damage. PLoS One. 2020, 15(2), e0229613. https://doi.org/10.1371/journal.pone.0229613.
Trapella, C.; Rizzo, R.; Gallo, S. et al. HelixComplex snail mucus exhibits pro-survival, proliferative and pro-migration effects on mammalian fibroblasts. Sci Rep 2018, 8, 17665. https://doi.org/10.1038/s41598-018-35816-3
Ballesteros-Sánchez, A.; Sánchez-González, J.-M.; Tedesco, G.R.; Rocha-de-Lossada, C.; Murano, G.; Spinelli, A.; Mazzotta, C.; Borroni, D. Evaluating GlicoPro Tear Substitute Derived from Helix aspersa Snail Mucus in Alleviating Severe Dry Eye Disease: A First-in-Human Study on Corneal Esthesiometry Recovery and Ocular Pain Relief. J. Clin. Med. 2024, 13, 1618. https://doi.org/10.3390/jcm13061618
Mencucci, R.; Strazzabosco, G.; Cristofori, V.; Alogna, A.; Bortolotti, D.; Gafà, R.; Cennamo, M.; Favuzza, E.; Trapella, C.; Gentili, V.; et al. GlicoPro, Novel Standardized and Sterile Snail Mucus Extract for Multi-Modulative Ocular Formulations: New Perspective in Dry Eye Disease Management. Pharmaceutics 2021, 13, 2139. https://doi.org/10.3390/pharmaceutics13122139
Comments 9: Overall the discussion is quite extensive, the main message is not clear and the back and forth turns it difficult to understand when the authors are talking about their own results or results from other studies.
Authors should make an effort to summarise and integrate the different results, discussing them as a whole whenever possible. Creating subheadings would also facilitate the understanding. What is the main narrative of the study? That is not clear by reading the discussion.”
Response 9: The Discussion section has been substantially revised; subheadings have been added. We have made an effort to address all the reviewer's comments. The text has been condensed, the results summarized and integrated, and subheadings have been presented for the convenience of the readers. We have made an effort to present the main emphasis of the study. Moreover, a large part of the sections Introduction, Results, and Discussion has been rewritten according to the comments of both reviewers. The entire manuscript was edited in English.
Round 2
Reviewer 1 Report
Comments and Suggestions for Authors
No 没有
Author Response
Dear Reviewer,
We would like to express our sincere gratitude for your second review of our manuscript and your positive evaluation.
Sincerely Yours:
Prof. Ekaterina Krumova, PhD
Department of Mycology
The Stephan Angeloff Institute of Microbiology
Bulgarian Academy of Sciences
Sofia, Bulgaria
e-mail: ekrumova@abv.bg
Reviewer 2 Report
Comments and Suggestions for Authors
The authors were careful in their corrections and have made substantial improvements, which are most clear in the discussion section. Overall, the study presents a good dataset that I believe will be of interest to a broad readership. I have some final minor comments that I would like authors to address:
1- As pointed out in the author’s response to my first round of comments I believe that authors agree that the mechanism of action was not determined. Determining mechanisms of action require much more in-depth investigation to establish a well-defined chain of events connecting molecules, receptors and organelles that explain in detail how a trigger leads to a specific response. This has not been done by the authors in the current study, as it's unknown whether ROS is cause or consequence and what are their origin. With that in mind, I believe authors must tone down some of their words, for instance, replacing “mechanism of action” by “mechanism of damage”. After all, there is no doubt oxidative stress is part of the mechanism of damage in their study, even if the mechanism of action has not yet been established. Therefore, the statement in the abstract “In addition, the role of oxidative stress in the mechanism of action was determined” should be adjusted to something like “In addition, oxidative stress was established as part of the mechanism of damage…” and the Title should also be adjusted to something like “Mechanisms of Damage of Oxidative Stress in Antifungal Activity of Two 2 Mollusc Fractions on Resistant Fungal Strains”. If adjustments like these ones are employed, then no conceptual mistakes will be made.
2- It could be just the PDF version I am revising, but Figures 3 and 4 have no letters to point differences in statistical significance between groups and in Figures 5 and 6 the letters are misplaced or absent in some cases.
3- In their response to my first comments the authors mentioned: “Our results also prove the lack of cytotoxicity of the mucus fraction on human skin fibroblasts (BJ) and human keratinocytes (HaCaT), which are shown in our new study "Synergistic antibacterial effect of the mucus fraction of C. aspersum with antibiotics against pathogenic bacteria isolated from wounds of diabetic patients", submitted to Antibiotics (MDPI).” I believe this is a very important point and a couple of lines should be added to the discussion (referencing their article if possible) to highlight the safety of these mucus fractions to humans.
Author Response
Dear Reviewer,
We appreciate all your valuable comments on our manuscript. We have revised the manuscript according to the comments and suggestions. We believe that the manuscript has been further improved. Here are our corrections.
General comments: The authors were careful in their corrections and have made substantial improvements, which are most clear in the discussion section. Overall, the study presents a good dataset that I believe will be of interest to a broad readership. I have some final minor comments that I would like authors to address:
Comments 1. As pointed out in the author’s response to my first round of comments I believe that authors agree that the mechanism of action was not determined. Determining mechanisms of action require much more in-depth investigation to establish a well-defined chain of events connecting molecules, receptors and organelles that explain in detail how a trigger leads to a specific response. This has not been done by the authors in the current study, as it's unknown whether ROS is cause or consequence and what are their origin. With that in mind, I believe authors must tone down some of their words, for instance, replacing “mechanism of action” by “mechanism of damage”. After all, there is no doubt oxidative stress is part of the mechanism of damage in their study, even if the mechanism of action has not yet been established. Therefore, the statement in the abstract “In addition, the role of oxidative stress in the mechanism of action was determined” should be adjusted to something like “In addition, oxidative stress was established as part of the mechanism of damage…” and the Title should also be adjusted to something like “Mechanisms of Damage of Oxidative Stress in Antifungal Activity of Two 2 Mollusc Fractions on Resistant Fungal Strains”. If adjustments like these ones are employed, then no conceptual mistakes will be made.
“Mechanisms of Damage of Oxidative Stress in Antifungal Activity of Two 2 Mollusc Fractions on Resistant Fungal Strains”.
Response 1. We agree and have replaced “mechanism of action” by “mechanism of damage” throughout the text.
The statement in the abstract “In addition, the role of oxidative stress in the mechanism of action was determined” has been changed to “In addition, the role of oxidative stress in the mechanism of damage was determined”.
The title of the manuscript does not include any judgment on OS as a mechanism of action of the mollusc fractions used: “The Role of Oxidative Stress in Antifungal Activity of Two Mollusc Fractions on Resistant Fungal Strains”. We believe that it is consistent with the reviewer's thesis on the role of OS only as a cause of the damage found. In this context, it need not be modified.
Comments 2. It could be just the PDF version I am revising, but Figures 3 and 4 have no letters to point differences in statistical significance between groups and in Figures 5 and 6 the letters are misplaced or absent in some cases.
Response 2. We thank the reviewer for this comment. The letters indicating differences in statistical significance between groups have been noted in all figures.
Comments 3. In their response to my first comments the authors mentioned: “Our results also prove the lack of cytotoxicity of the mucus fraction on human skin fibroblasts (BJ) and human keratinocytes (HaCaT), which are shown in our new study "Synergistic antibacterial effect of the mucus fraction of C. aspersum with antibiotics against pathogenic bacteria isolated from wounds of diabetic patients", submitted to Antibiotics (MDPI).” I believe this is a very important point and a couple of lines should be added to the discussion (referencing their article if possible) to highlight the safety of these mucus fractions to humans.
Response 3. Thank for your correct comments. To the section Discussion a passage pointing our results highlighted the safety of used mucus fractions to humans have been added (p. 15, lines 689-595).